# Origin of outer tropical cyclone rainbands

Cheng-Ku Yu [1] ✉, Che-Yu Lin[1] & Chi-Hang Pun [1]

Outer tropical cyclone rainbands (TCRs) are a concentrated region of heavy precipitation and hazardous weather within tropical cyclones (TCs). Outer TCRs pose considerable risk to human societies, but their origin remains unresolved. Here, we identify a total of 1029 outer TCRs at their formative stage from 95 TCs and present a large collection of radar observations in order to establish a robust foundation of the natural diversity of rainband origin. The results show the dominance of outer origin for the observed outer TCRs, in distinct contrast to theoretical modeling works of outer TCRs, which propose inner-origin scenarios. Our analysis also suggests that squall-line dynamics are a common, but not the sole, mechanism responsible for outer TCR formation. The nature of preexisting outer precipitation is found to be an important factor to influence the squall-line and non-squall-line outer TCR initiation.

Tropical cyclones (TCs) are the most devastating storms in nature since they can result in the significant loss of human life and tremendous property damage due to torrential rainfall and intense winds[1,2]. In addition to eyewalls, tropical cyclone rainbands (TCRs) are a primary, concentrated region of heavy precipitation and hazardous weather within TCs[3–10]. TCRs not only play a vital role in influencing the development and intensity of TCs[11–18] but also represent vast threats to societies because of their high potential to produce catastrophic rainfall and floods[19–23]. Therefore, improving our fundamental understanding of how TCRs form and develop is critically important for disaster mitigation and the prediction of the natural hazards and climate extremes in TC-prone regions.

TCRs are conventionally classified into inner and outer rainbands based on their location relative to the TC center. Inner TCRs are located within the TC inner core that has a radial distance of 100–200 km or ~3 times the radius of maximum wind (RMW), whereas outer TCRs are located beyond the inner core[17,24]. Moist convection of inner TCRs is usually weaker and organized into a more circular geometry due to the pronounced influence of the inner-core vortex and filamentation effect[5,8,25]. The precipitation features of outer TCRs are typically asymmetric but much more convective than those of inner TCRs due to a larger convective instability in the outer region of TCs[5,8,17,26–31]. Outer TCRs tend to have stronger and broader impacts on human and natural communities through the generation of severe weather conditions and flooding[21,28,29,31,32].

Although the processes that form the inner TCRs have not been wholly conclusive[33], a general consensus is that their appearance is related to the inner-core wave activities (e.g., vortex Rossby waves, VRWs) initiated near the eyewall[34–38]. However, the origin of the outer TCRs remains uncertain and the formative mechanisms proposed previously are diverse. Earlier modeling and theoretical investigations have suggested that outer TCRs are probably a manifestation of outward-propagating gravity waves (GWs) that are excited close to the TC center or inside the inner core[39–42]. In this view, the moist convection occurring in outer TCRs may be triggered by gravity-wave-induced upward motions[42,43]. Despite the potential discrepancy in the propagation speed between theoretical GWs and observed outer TCRs[3,30,42–47], what we learned from these previous works is still unable to adequately address whether outer TCRs would be initially triggered by the GWs within the inner-core region and then propagate outward to the outer region of TCs.

Recently, the inner-origin scenario above was further explored by an idealized numerical TCR simulation by Li et al.[48]. That study demonstrates that convectively active outer TCRs form as weakly convective inner TCRs propagate to the outer region where filamentation and stabilization are reduced. Li et al.[48] argue that their simulated inner TCRs would most likely be related to inner-core VRWs instead of GWs. In the context of the inner origin, the inner and outer TCRs are supposed to have no inherent distinction, although their associated precipitation characteristics may undergo some sort of convective transformation in response to the differences in environmental conditions between the inner and outer regions of TCs.

Although many previous theoretical and modeling works have emphasized the importance of the inner-origin processes for outer TCRs, they also seem likely to form locally near the inner-core boundary or in the outer region of TCs, which may be considered outer-origin. Understanding of the outer-origin processes has been

[1]Department of Atmospheric Sciences, National Taiwan University, Taipei, Taiwan. ✉e-mail: yuku@ntu.edu.tw

limited due to scarce observations during the formative stage of outer TCRs. A particular, well-known type of rainband usually exists near the boundary between the inner-core vortex of TCs and environmental flow. This type of rainband has been referred to as the "principal band (PB)"[5] and is typically quasi-stationary relative to the TC center[5,7,8,49,50]. Outer TCRs are a broader class of rainbands than just PB that exhibits unique precipitation and kinematic features tied to TC vortex dynamics. Although possible mechanisms responsible for the formation of PB have been explored by numerical modeling[51], previous PB observational studies rely dominantly on the analyses of airborne flight-level and radar data collected over a relatively short period of time during the mature or late stage of oceanic TCRs. Hence, these observational works do not provide strong clues to where and how outer TCRs form within TCs.

With advances in high-resolution radar and surface observations and numerical simulations, a growing number of TC studies have improved our understanding of the detailed structural characteristics of outer TCRs[28–31,44,45,52–56]. These investigations have revealed the frequent presence of surface cold pool signatures and squall-line-like airflow patterns for outer TCRs, which imply the potential importance of squall-line dynamics on the development of moist convection associated with outer TCRs. However, none of the previous observational studies, except for the TCR case described in Yu et al.[30], capture the formative stage of outer TCRs or confirm how relevant the squall-line dynamics would be in the outer TCR initiation process.

To the best of our knowledge, Yu et al.[30] is the only observational study of TCRs to provide a complete picture of the outer-origin processes. By tracking the history of a long-lived outer TCR associated with Typhoon Jangmi (2008), Yu et al.[30] documented that the outer rainband was initiated as it detached from the upwind segment of a stratiform precipitation region located slightly beyond the inner-core boundary. A distinct transformation from generally stratiform precipitation to highly convective precipitation, which was related to the

increasing intensity of convectively generated cold pools and their interactions with low-level vertical shear[57], was documented as the rainband propagated cyclonically outward.

The diurnal forcings and large-scale vertical wind shear have been both recognized to have a strong impact on the development and structure of TC convection. Previous studies have provided observational evidence that these two factors can influence convective intensities and distributions over a very broad area not only within the inner region of TCs but also beyond their inner core[58,59]. It is thus likely that the outer TCR origin would be also modulated by diurnal and shear effects. However, this likelihood has never been confirmed, and whether the formation of outer TCRs would exhibit diurnal cycles or some sort of dependence on large-scale vertical shear remains unclear.

A final possible contributor to the origin of outer TCRs, as proposed by a few earlier studies, is related to the boundary-layer instability from an Ekman-like flow[4,60–62]. On the basis of theoretical approaches, this kind of instability can occur in the rotational environment under the presence of suitable vertical shear in the boundary layer, taking the form of spiral-shaped roll vortices. Two of the most remarkable aspects for the instability-generated perturbations are their storm-relative stationary characteristics and a nearly constant crossing angle of about ~15° with respect to the TC center[4]. The crossing angle is defined as the angle between the rainband's orientation and the tangent to a circle with radius from the band to the TC center[3,4]. Different wavelengths of roll vortices, ranging from sub-kilometers to several kilometers, have also been observed in the tropical cyclone boundary layer[63–65]. However, the potential relevance of the boundary-layer instabilities and roll vortices to the appearance of outer TCRs remains highly uncertain at present due to the lack of observational validation.

It is clear from the literature reviews above that current research is still far from adequate to clarify the relative prevalence of the inner-origin and outer-origin scenarios and of various possible realistic formative scenarios for outer TCRs. To establish a robust foundation of knowledge regarding the natural diversity of the rainband origin, a critical and challenging task is thus to collect detailed observational information from a very early stage of rainband development from a considerable number of outer TCR cases.

The primary objective of this study is to use long-term radar observations collected during 2002–2019 to identify the origin of numerous outer TCRs associated with TCs as they approach the ocean areas near Taiwan. The island of Taiwan is located within the main path of TCs that originate over the northwestern Pacific Ocean, a basin with the highest TC activity on Earth[1]. The statistics based on the typhoon database provided by the Taiwan Central Weather Bureau (CWB) (https://rdc28.cwb.gov.tw/TDB/public/warning_typhoon_list/) indicate that, since 1958, there have been approximately four to ten TCs that approach the area around Taiwan every year. In addition, an excellent ground-based Doppler radar observation network has been in operation in Taiwan since the 2000s. Both of the above aspects make it possible to collect a large dataset of outer TCR cases as TCs approach from the northwestern Pacific. The particular focus of this study is to explore the important characteristics of outer TCRs during their initiation stage in terms of their formative location and attendant propagation. The observations consisting of 1029 outer TCR cases associated with 95 TCs can provide important insight into the origin of outer TCRs.

## Results

### Formative location and dynamic regime

A plan view of the formative locations, with respect to the typhoon center and the direction of typhoon motion, for all identified TCRs is illustrated in Fig. 1. A diverse horizontal distribution of TCR formation over a broad region is seen, with the majority of the TCRs located

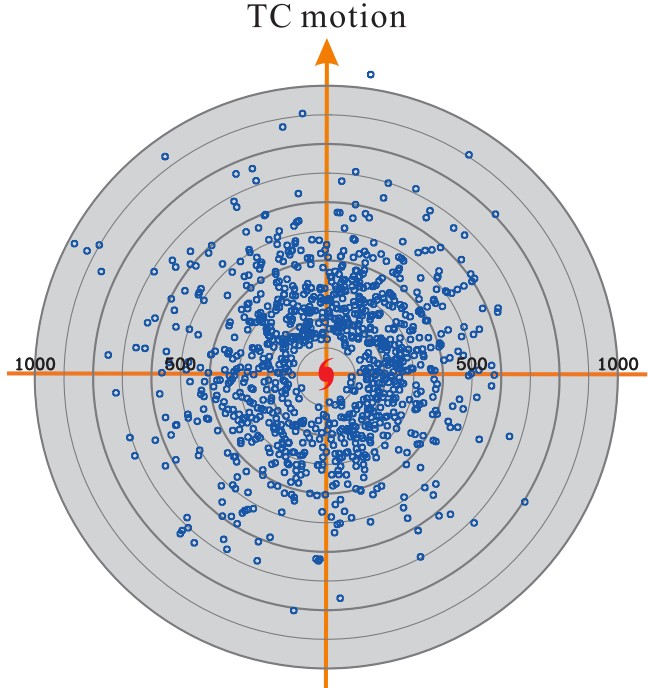

**Fig. 1 | Formative distribution of outer tropical cyclone rainbands (TCRs) within tropical cyclones (TCs).** Plan view of formative locations for all outer TCRs (1029 cases) identified in this study relative to the TC center and motion. The TC motion is toward the top of the page. The range rings are indicated every 100 km. Source data are provided as a Source Data file.

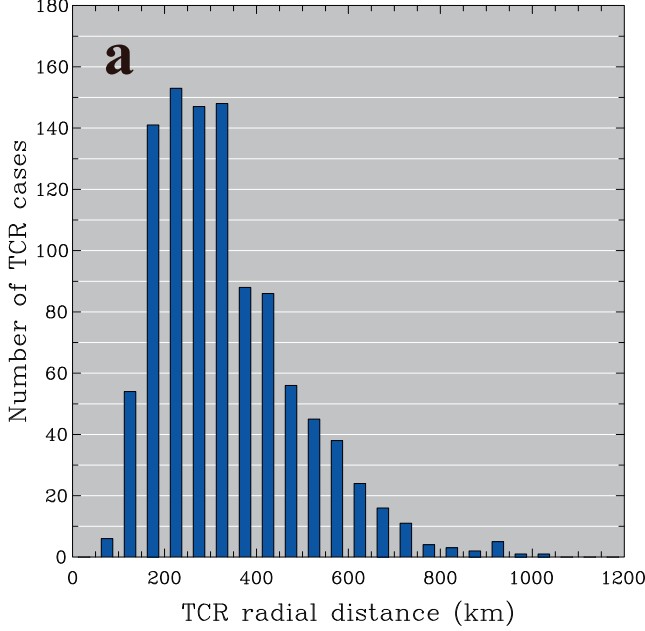

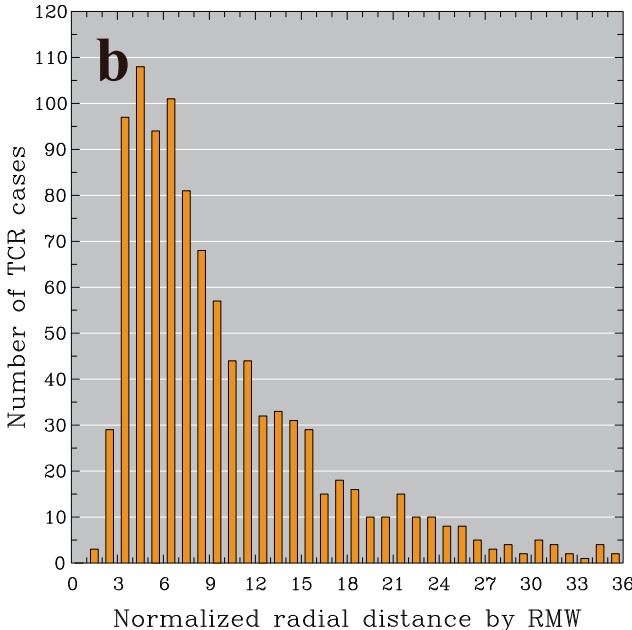

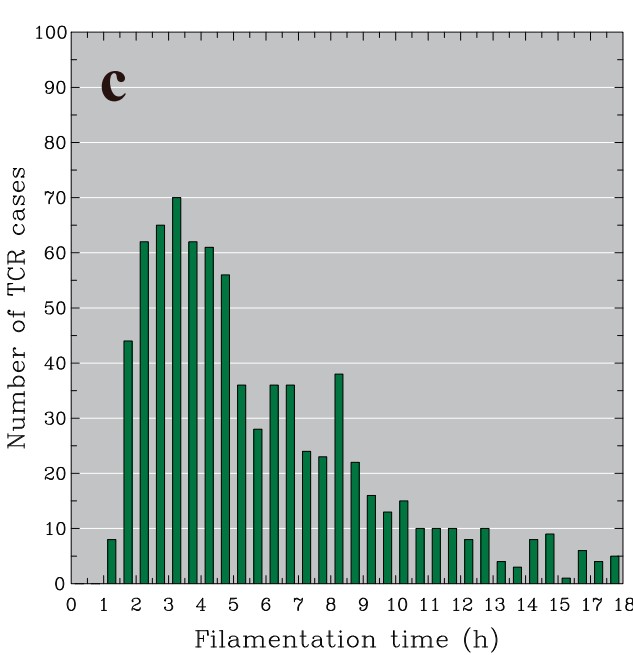

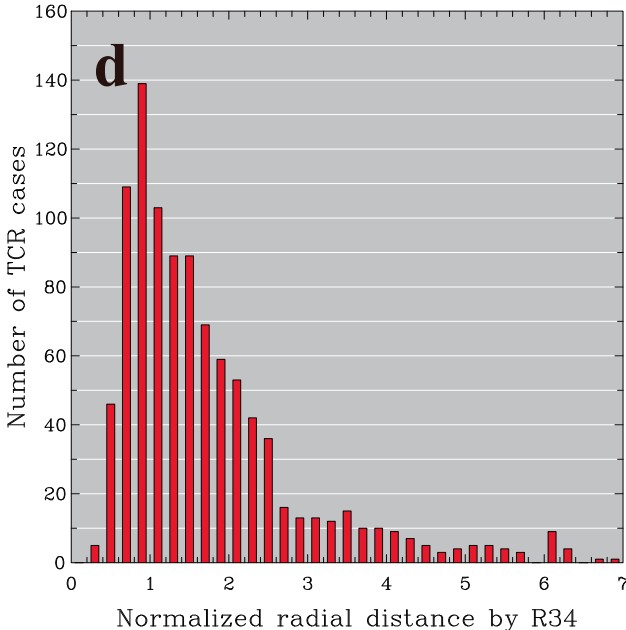

**Fig. 2 | Formation of outer tropical cyclone rainbands (TCRs) at different spatial and dynamic intervals.** Formative number of the observed outer TCR cases at different intervals of (**a**) radial distance (km), (**b**) normalized radial distance by the radius of maximum wind (RMW), (**c**) filamentation time (h) and (**d**) normalized radial distance by the radius of 34 kt winds (R34) provided by the Joint Typhoon Warning Center (JTWC). Source data are provided as a Source Data file.

within a radial distance of ~400 km. Nearly all TCRs are initiated beyond a radius of ~100 km. There are slightly more rainbands (~59% of the identified TCRs) in the front quadrants. This azimuthal variation of the outer TCR formation, however, would be due in part to the contamination by the inherent limitation of radar observations[9,29]. Specifically, the precipitation information of the front quadrants for a typical westward-moving TCs over the northwestern Pacific Ocean could be usually monitored by coastal radars in Taiwan for a longer period of time. In contrast, when the rear quadrants of TCs reach the observational coverage of coastal radars, the inner-core circulations of TCs have been closer to the landmass of Taiwan so both TC circulations and rainbands have greater chances to experience orographic

modifications and/or make landfall. This sampling preference of coastal radars is expected to capture more outer TCR cases in the front quadrants than in the rear quadrants, consistent with the asymmetric characteristic of formative locations of the observed outer TCRs, as shown in Fig. 1.

The formative number of TCRs as a function of radial distance, as shown in Fig. 2a, indicates that the most concentrated regions of TCR formation are between 150 and 350 km from the TC center, and 589 rainbands (~57%) are initiated within this radial interval. Nevertheless, considerable TCRs are observed to form in far outer TC regions beyond the radius of 350 km. The formative number of TCRs expressed as a function of normalized radial distance by RMW, as

shown in Fig. 2b, shows that a dominant portion of the identified TCRs (~97%) are initiated beyond the inner-core region (i.e., a distance of three times the RMW). Relatively higher formative numbers exist in the radii of 3–8 times the RMW, which accounts for ~47% of the identified TCRs. Peak TCR formation (108 cases) is located between 4 and 5 times the RMW. Outer TCRs that form in the outer vicinity adjacent to the inner-core boundary were also identified in a case study by Yu et al.[30]. Note that few of the outer TCRs (32 cases) are initiated inside three times the RMW (Fig. 2b), implying that the formation of these rainbands is related to the transition from outward-propagating inner TCRs to outer TCRs[48]. However, such an inner origin is not a frequent or dominant scenario for outer TCR formation because of a very low fraction of initiation within the inner-core region (only ~3% of the identified TCRs). It is fairly reasonable to conclude that actual outer TCRs tend to be initiated locally in the outer TC region (i.e., outer origin) and do not preferably originate from the inner-core region.

A complementary view of dynamic regimes for outer TCR formation can be provided by analyzing the filamentation time for each of the identified TCRs. As demonstrated in Rozoff et al.[25], the inner-core flow of TCs tends to be strain-dominated and is characterized by a rapid filamentation zone with a time scale less than that of deep moist convective overturning (~30 min). Relative magnitudes of the filamentation time are useful for providing a dynamic distinction between the inner and outer regions of TCs[25,66]. In this study, the filamentation time, as defined in Rozoff et al.[25], is calculated using the wind information at the formative location of each rainband at 1.5 km above mean sea level (MSL) from the ERA5 reanalysis data. The reason why the 1.5-km level is chosen for calculation is because this height is immediately above the typical top of TC boundary layer (0.5 ~ 1 km MSL) so the winds from this height are expected to better represent the TC intensity and circulation. Figure 2c indicates that outer TCR formation occurs in regions with filamentation times greater than 1 h. None of the outer TCRs is initiated within the rapid filamentation zone (i.e., filamentation time <30 min). The dominant preference of outer origin for the observed rainbands supports what we see in Fig. 2b.

Since outer TCR formation is observed to be active over extensive outer regions of TCs, as shown in Figs. 1, 2a–c, one intriguing question emerges, what is the degree of prevalence for outer TCRs to form at radii larger than the storm radius of TCs? The size of a TC has been practically defined as the radius of a certain threshold of near-surface wind speed or by the radial extent of the outermost closed isobar[67–69]. Because the TC size may vary considerably with different thresholds chosen, we use the average radius of 34 kt winds (R34) as a storm radius for TCs in this study. R34 is commonly considered as the area of the TC warnings issued by operational forecasting centers[31]. The statistical distribution of the TCR formative number, as a function of normalized radial distance by JTWC-recorded R34, is shown in Fig. 2d. The peak outer TCR formation (139 cases) is observed within the interval of normalized radial distances between 0.75 and 1, which is close to the outer boundary of the storm radius of R34. Although the formative numbers tend to decrease generally with increasing radial distance in regions beyond the storm radius, the formative fractions of outer TCRs inside and outside the storm radius are dramatically different, which are calculated to be 29% and 71%, respectively. These statistics further indicate that the majority of outer TCR formation actually occurs in far outer regions beyond the operational TC alert area. This finding is impressive and suggests an essential need to expand the area of routine TC alerts, particularly given the frequent development of severe weather conditions associated with outer TCRs[29,31].

It is noteworthy that diurnal forcings, as described in the Introduction, are well known to influence convective development of TCs.

The formative times of all outer TCRs observed in this study are also analyzed to understand if there are any diurnal signals. It is found that the number of outer TCRs formed at different times in a day is similar and approximately equal to 80 cases for each time interval. This result generally reflects the lack of obvious diurnal variations for the formation of the studied outer TCRs.

**Propagation and geometry characteristics**

The large collection of TCR cases in this study provides a good opportunity to explore the generality and diversity of propagation and geometry characteristics for outer TCRs near Taiwan. Statistical results of the radial and tangential propagation velocities and the rainband's width and crossing angles for all identified TCRs are shown in Fig. 3. The identified TCRs possess significant variabilities in the radial propagation speed, but the dominant majority (964 TCR cases, ~94%) are within the range of ±10 m s$^{-1}$ (Fig. 3a). Outward propagation (i.e., positive radial velocities) tends to be more frequent (660 TCR cases, ~64%), whereas inward propagation is also evident from a considerable number of outer TCRs (369 TCR cases, ~36%). This statistical finding is significant and indicates that outward propagation is common but not dominant for outer TCRs. Whether these propagation characteristics are consistent with wave theories will be explored later in this section. The peak TCR number exists in the velocity interval of 2–4 m s$^{-1}$, which is roughly comparable to the outward propagation speed of several outer TCRs (3–7 m s$^{-1}$) reported in previous observational studies of TCs[3,30,43]. Small proportions of the observed outer TCRs have slow radial propagations within ±1 m s$^{-1}$ (Fig. 3a), consistent with the quasi-stationary characteristics of PB[5] and spiral-shaped disturbances generated by boundary-layer instabilities in a theoretical framework[61]. However, these rainbands are few (only 144 cases, ~14%), and the radially propagating feature is prevalent from our TCR dataset. Assuming the 2–5 times RMW as an approximate area for the outer boundary of the inner-core vortex of TCs, where PBs typically form, only 13 out of 144 outer TCR cases are actually located downshear within this area. Such a small number of TCR cases (~1% of all identified outer TCRs) suggests that the possible impact of PB-like rainbands on the outer TCR dataset analyzed in this study may be considered negligible. One of the primary factors contributing to the potential preclusion from the class of outer TCRs is related to the fact that PBs tend to occur preferentially inside the TC vortex core or near its outer boundary (~3 times RMW), as learned from the comprehensive examination of radar-observed precipitation field associated with 95 TCs investigated in this study.

Compared with the radial propagation, the tangential propagation of outer TCRs is generally much faster and predominantly cyclonic (i.e., positive tangential velocities) (Fig. 3b), which is consistent with the influence of TC vortex circulations. The mean tangential propagation for all identified TCRs is calculated to be 10.4 m s$^{-1}$, which is somewhat slower than the average ambient tangential flow (14.4 m s$^{-1}$) within the layer of precipitation associated with the rainbands (0.1 ~ 12 km MSL).

Most of the observed outer TCRs (748 cases; 73%) have cross-band widths ranging from 5 to 25 km (Fig. 3c), equivalent to a horizontal wavelength of 10 to 50 km, as approximated by twice the radar-observed TCR width. These rainband scales are much larger than the boundary-layer roll vortices, with typical horizontal wavelengths on the order of a few hundreds of meters to several kilometers[62–65]. The most frequent occurrence of the observed crossing angles is found between 10° and 30° (Fig. 3d), which is consistent with typical values of 20° to 25° reported in the literature[4]. However, a wide distribution of crossing angles, such as the evidence of larger and negative values for some observed rainbands, implies complexities in the TCR structures and dynamics.

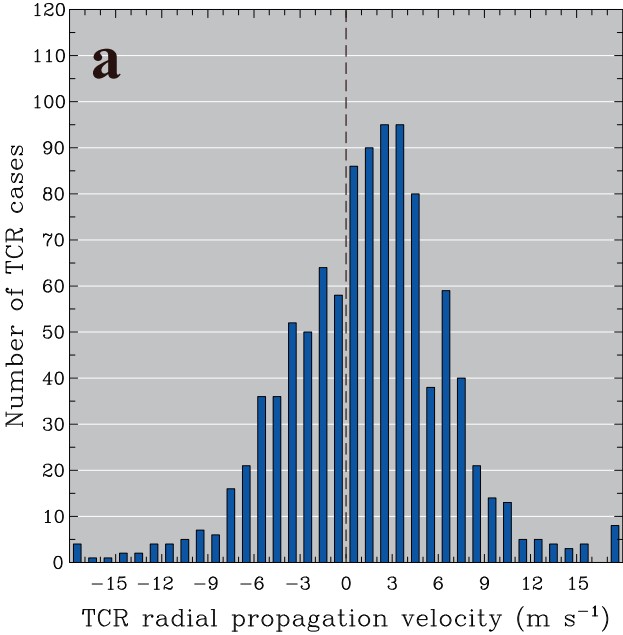
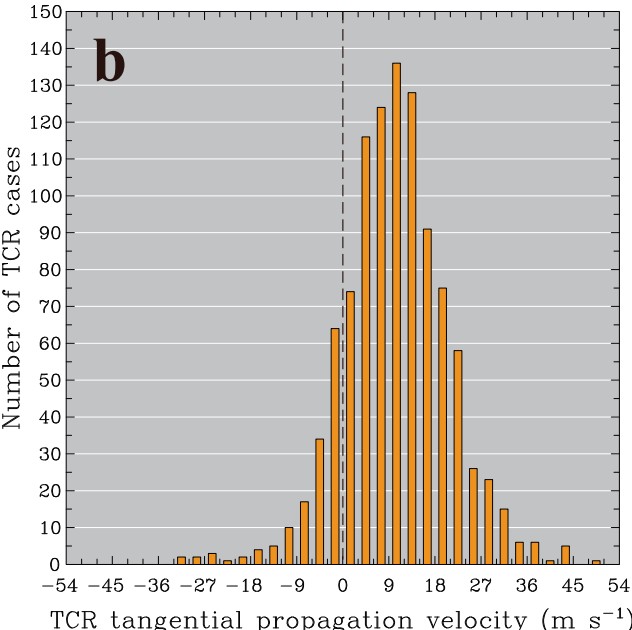
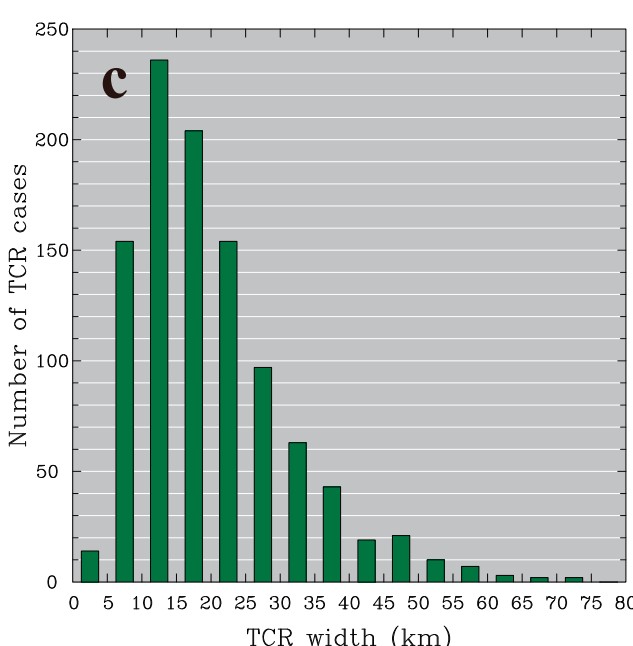
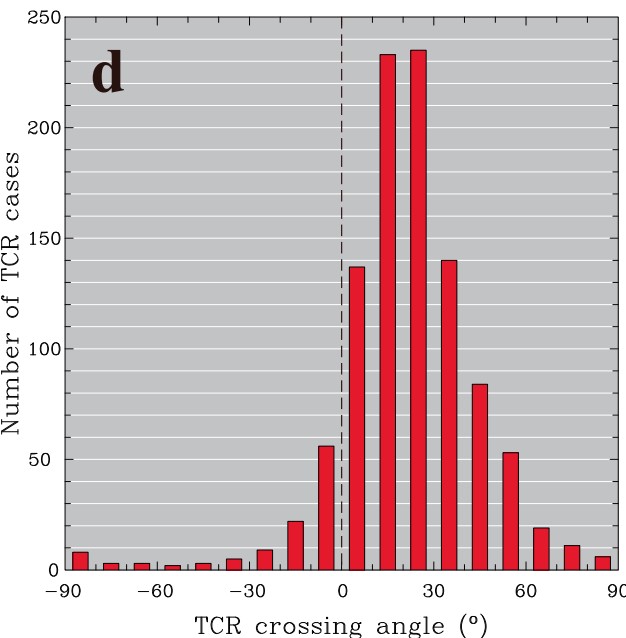

**Fig. 3 | Number of outer tropical cyclone rainbands (TCRs) as a function of rainband propagation and geometry.** Number of observed outer TCR cases at different intervals of (**a**) radial propagation velocity (m s⁻¹), (**b**) tangential propagation velocity (m s⁻¹), (**c**) rainband width (km) and (**d**) rainband crossing angle (°). Vertical lines in (**a**), (**b**) and (**d**) denote the zero value along the horizontal axis. Source data are provided as a Source Data file.

To evaluate the relevance of wave activities to the initiation of outer TCRs, the theoretical phase velocities for two well-known, important wave types (GWs and VRWs) active within TCs are calculated and compared with the observed propagation velocities of outer TCRs presented above. The classical dispersion relation of internal gravity waves may be written as[46,70]:

$$C_{gw} = \frac{\upsilon}{k} = \pm \frac{N}{\sqrt{k^2 + m^2}} \qquad (1)$$

where $C_{gw}$ is the GW phase velocity, $\upsilon$ is the intrinsic wave frequency, $k$ is the horizontal wave number, $m$ is the vertical wave number, and $N$ is

the Brunt-Väisälä frequency that is computed from the ERA5 reanalysis data below 12 km MSL nearest the location of each identified TCR. The value equal to twice the radar-observed TCR width is again used to approximate the horizontal wavelengths. The vertical wavelength is assumed to be 12 km MSL based on the typical depth of convective motions associated with outer TCRs[29]. The scatterplot shown in Fig. 4a indicates the lack of consistency between the observational TCR radial propagation velocities and theoretical GW phase velocities. Most of the GW phase velocities are confined to a narrower range at approximately ±20 m s⁻¹, which is the same order of magnitude as the typical radial propagation speeds of several meters per second for GWs excited within TCs[42,44–47]. These theoretical propagation estimates

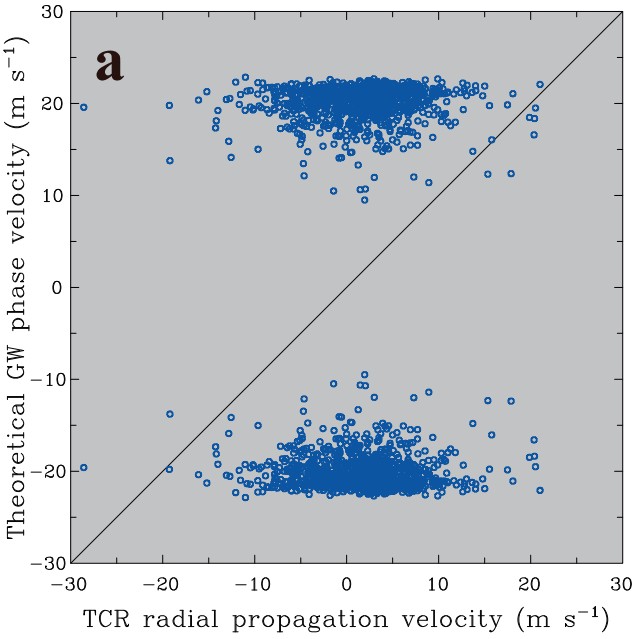

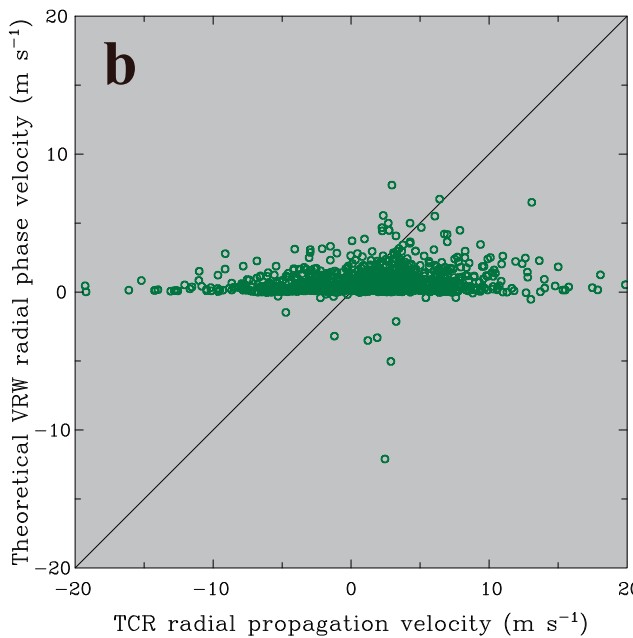

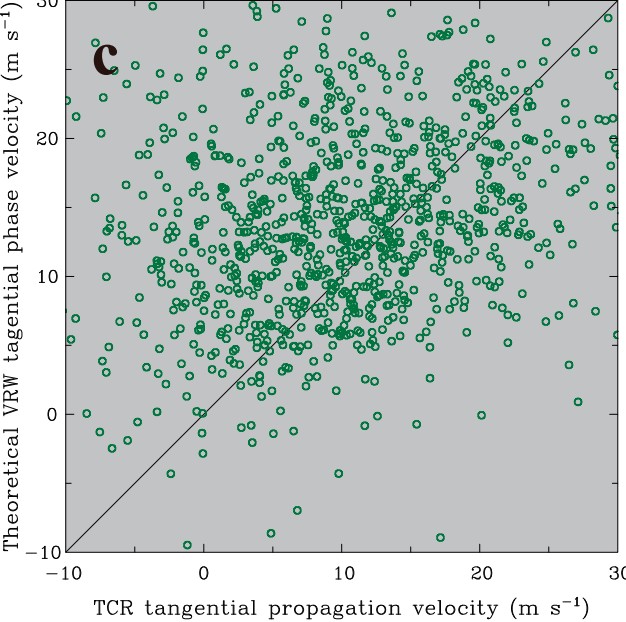

**Fig. 4 | Comparisons of observed rainband propagations with gravity wave (GW) and vortex Rossby wave (VRW) phase velocities. a** Scatterplot of the observed radial propagation velocities vs. the theoretically predicted GW phase velocities for all identified outer TCR cases. **b** Scatterplot of the observed radial propagation velocities vs. the theoretically predicted VRW radial phase velocities for all identified outer TCR cases. **c** Same as in (**b**) but showing the observed tangential propagation velocities vs. the theoretically predicted VRW tangential phase velocities. In each panel, the diagonal line representing perfect agreement between the observational and theoretical propagation velocities is also indicated for reference. Source data are provided as a Source Data file.

are distinctly separated from the primary velocity distributions of the observed radial propagation, which are within the range of $\pm 10\,\mathrm{m\,s^{-1}}$ (Figs. 3a, 4a).

The theoretical VRW frequency ($\omega$) and radial and tangential phase velocities ($C_r$ and $C_\lambda$), as explored by Möller and Montgomery (2000)[71], may be expressed as:

$$\omega = n\bar{\Omega} + \frac{n}{R}\frac{\bar{\xi}}{\bar{q}}\frac{\frac{\partial \bar{q}}{\partial r}}{\left[k^2 + \frac{n^2}{R^2} + \frac{\bar{\eta}\bar{\xi}m^2}{N^2}\right]},\tag{2}$$

$$C_r = \frac{\omega}{k}\tag{3}$$

$$C_\lambda = \frac{\omega R}{n}\tag{4}$$

where $k$, $n$, and $m$ are the radial, tangential, and vertical wave numbers, respectively; $R$ is the reference radius; $N$ is the Brunt-Väisälä frequency; $\bar{\Omega}$ is the angular velocity; $\bar{\xi}$ is the inertia parameter; $\bar{\eta}$ is the absolute vorticity; and $\bar{q}$ is the potential vorticity. The last three variables are

estimated by using the formulation of Möller and Shapiro[72]:

$$\bar{\xi} = f + \frac{2\bar{V}}{R} \qquad (5)$$

$$\bar{\eta} = f + \frac{1}{R}\frac{\partial(R\bar{V})}{\partial r} \qquad (6)$$

$$\bar{q} = \frac{1}{\bar{\xi}}\left[\bar{\eta}\bar{\xi}N^2 - \bar{\xi}^2\left(\frac{\partial V}{\partial z}\right)^2\right] \qquad (7)$$

where $V$ is the tangential wind; $\bar{V}$ is the mean tangential wind; $f$ is the Coriolis parameter; and $N$ is the Brunt-Väisälä frequency. As described earlier, the radial wavelength ($k$) and the vertical wavelength ($n$) are again approximated as being twice the radar-observed TCR width and 12 km, respectively. The mean tangential wind in (5) and (6) is calculated by averaging tangential winds within the layer of 0.1–12 km MSL that is corresponding to the vertical wavelength of considered wave motions. The tangential vertical shear in (7) is approximated by the difference in the tangential wind between 0.1 and 12 km MSL. The reason why we choose the 0.1 km MSL as the lowest level in these calculations is that this height is well above the top of surface layer so the uncertainty on winds due to strong and direct impact of surface friction can be appropriately eliminated. A typical tangential wave number ($n = 2$) that characterizes VRWs is assumed[37,73]. Note that the tangential wave number does not have a strong impact on the calculated propagation velocities, and choosing other tangential wave numbers, such as $n = 1$ or 3, for calculation in (2) gives rise to nearly identical propagation estimates. The radial distance of the observed TCRs is used to represent the reference radius ($R$) and $\bar{\Omega} = \bar{V}/R$. The scatterplot of the observed TCR propagation velocities versus theoretical VWR phase velocities reveals quite diverse distributions, with low correlation coefficients equal to 0.1 and 0.2 for the radial and tangential velocities, respectively (Fig. 4b, c). In particular, the VRW theory predicts generally slower ($<-3\,\mathrm{m\,s^{-1}}$) and predominantly outward propagations, in contrast to a very wide range of the observed velocities with evidence of both outward and inward propagation (Fig. 4b). Although some of the outer TCRs tend to have good agreement between observations and theory (i.e., lying closely along the diagonal line), the fraction of these TCR cases is small. For example, only 5.5% of the observed TCRs (i.e., 57 TCR cases) have magnitudes of differential velocity vectors between observational and theoretical propagations less than $2\,\mathrm{m\,s^{-1}}$. Indeed, the mean magnitude of the differential vectors calculated from all of the outer TCRs identified in this study can be as large as $10.6\,\mathrm{m\,s^{-1}}$. The statistical analyses provide evidence for a large discrepancy in propagation characteristics between the observed outer TCRs and theoretical GWs/VRWs. These results are considered robust, and suggest that the GW/VRW disturbances have little impact on the origin of outer TCRs.

## Squall-line dynamics

As described in the Introduction, nearly none of the previous TC investigations are capable of clarifying the degree to which squall-line dynamics may contribute to the origin of outer TCRs. Moreover, the limited number of outer TCRs documented previously is also far from adequate to address how prevalent the squall-line dynamics would operate during the initiation process for outer TCRs. With a large dataset of outer TCRs collected during their formative stage in this study, it is now possible to evaluate the likelihood.

According to the classical squall-line theory proposed by Rotunno et al.[57], the band-normal vertical shear toward the downshear side (i.e., frontward shear) and the warm, moist inflow (i.e., band-relative front-to-rear flow) present at low levels are two of the most fundamental characteristics of squall-line dynamics. The presence of frontward

shear is not only the most typical environment characterizing squall-line systems but is also the mandatory criterion needed to facilitate the lifting of convectively generated cold pools[57,74]. The low-level front-to-rear flow that feeds warm air to the leading edge of squall lines can be considered a critical source of convective energy for the development of deep convection. If squall-line dynamics plays a substantial role in the initiation of outer TCRs observed in this study, one might expect to see some sort of preference in the distribution of the low-level band-normal vertical shear and band-relative flow consistent with the above outlined squall-line characteristics.

Using the TCR's propagation, orientation and position information and the ERA5 reanalysis data, a scatterplot of the low-level band-normal vertical shear vs. the band-relative flow for all outer TCR cases can be constructed, as shown in Fig. 5a. In this analysis, the low-level vertical shear is approximated by the difference in the band-normal wind between 0.1 and 2 km MSL. This layer is corresponding to the mean depth of cold pools associated with outer TCRs, as revealed by previous studies of TCs[28–30]. Similar to the consideration for the VRW calculations as described in the previous section, choosing the lowest level at 0.1 km MSL is to preclude the uncertainty on winds due to direct impact of surface friction. Moreover, the band-normal wind is calculated below 1.5 km MSL, corresponding to the mean depth of low-level inflow feeding the leading edge of the outer TCRs documented in this study. A large proportion of the outer TCRs (682 cases; 66%) is observed within the fourth quadrant (IV) of the analysis with positive band-normal vertical shear (i.e., frontward shear) and negative band-normal wind components (i.e., the front-to-rear flow) (Fig. 5a). There are relatively fewer TCR cases, with 20%, 9% and 5% observed in other quadrants (I, II and III). Interestingly, the percentage of outer TCR cases (i.e., 66%) that exhibit squall-line environmental characteristics is close to the degree of prevalence of outer TCRs (58%) with squall-line-like airflow structures reported in Yu et al.[29]. This striking environmental preference suggests that the squall-line dynamics is likely to operate and contribute to rainband initiation for a significant portion of the observed outer TCRs. However, it should be noted that a squall-line environment, as discussed herein, is not a sufficient condition but is merely a necessary condition to judge the relevance between outer TCRs and squall lines because the presence of the squall-line environment does not fully guarantee that squall-line dynamics is occurring.

Some clarification on the issue above may be performed by evaluating the relative consistency of whether the observed outer TCRs propagate at a speed similar to the atmospheric cold pool. In this evaluation, we simply classify the observed outer TCRs, according to their environmental characteristics, into the squall-line TCR group (i.e., 682 TCR cases seen in quadrant IV of Fig. 5a) and the non-squall-line TCR group (i.e., 347 TCR cases seen in quadrants I, II and III of Fig. 5a). The velocity of an atmospheric cold pool ($V_{cp}$) perpendicular to its leading edge can be approximated by the following expression[75]:

$$V_{cp} = k\left[\frac{gH(\theta_{v2} - \theta_{v1})}{\theta_{v2}}\right]^{\frac{1}{2}} + bV_c \qquad (8)$$

where $H$ is the depth of the cold air, $\theta_{v2}$ and $\theta_{v1}$ are the ambient and cold pool virtual potential temperatures, respectively, $V_c$ is the ambient band-normal wind component ahead of the cold pool (negative for wind moving toward the cold air and vice versa), and $k$ and $b$ are empirical constants. The best fit of (8) to atmospheric observations gives constant $k$ and $b$ values equal to 0.9 and 0.6, respectively[75]. Considering the propagation of cold pools embedded within the moving TCs, we may need to approximate $V_c$ in (8) by taking the TC motion into account so that the positive and negative $V_{cp}$ calculated thus represent band-normal propagation motions away from and toward the TC center, respectively. Since direct measurements for the depth of the cold air and the cold pool virtual potential temperatures during the TCR initiation are not available, we attempt to determine

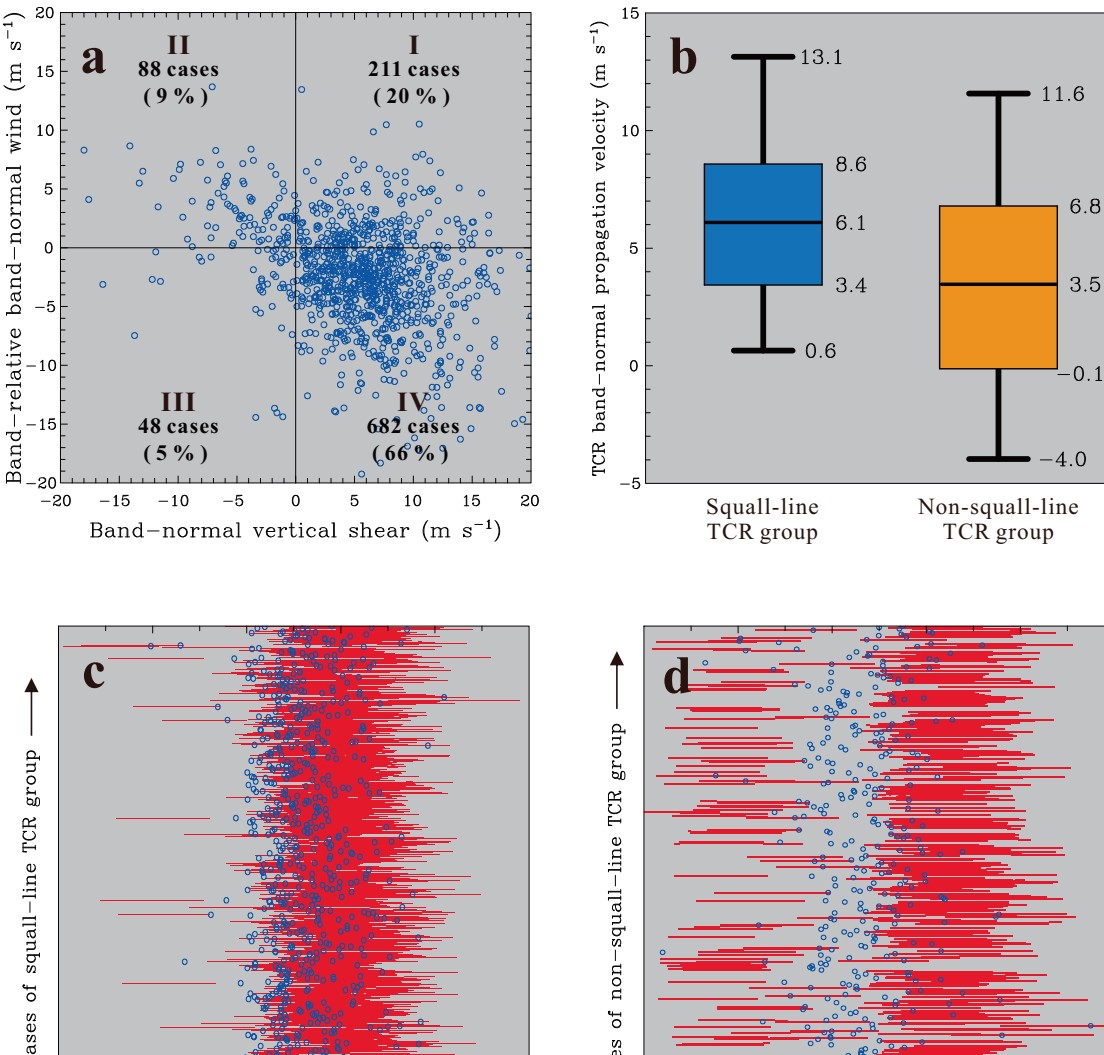

**Fig. 5 | Evaluation of squall-line dynamics based on environmental factors and rainband propagation characteristics. a** Scatterplot of the low-level band-normal vertical shear vs. the band-relative band-normal wind component for all outer tropical cyclone rainband (TCR) cases identified in this study. The low-level vertical shear and band-relative flow are calculated and averaged below 1.5–2 km MSL (see details in the text) from ERA5 reanalysis data nearest the formative location of each outer TCR case. The number of outer TCR cases and the corresponding percentage observed within each quadrant are also indicated. **b** The box and whisker plot of the observed cross-band propagation velocities (m s⁻¹) relative to the tropical cyclone

(TC) motion for the squall-line and non-squall-line TCR groups. Boxes denote 25th–75th percentiles (the interquartile range), with a horizontal line at the mean value. Vertical thick lines (whiskers) extend from the 5th to 95th percentile values. **c** Observed cross-band propagation velocities relative to the TC motion (small blue circles) and theoretically predicted velocity ranges of cold pools (red shading) for each of the cases of the squall-line TCR group. Positive propagation velocities represent motions away from the TC center, and vice versa. **d** Same as in (**c**) but showing the results for the non-squall-line TCR group. Source data are provided as a Source Data file.

the velocity range of $V_{cp}$ by giving a possible range of values for $H$ and the cold pool intensity (i.e., $\theta_{v2}$ -$\theta_{v1}$) and to check if the propagation velocity of the observed outer TCRs fit well into the calculated velocity range of $V_{cp}$. Based on previous observational studies of outer TCRs[28–30], the most reasonable ranges of values for $H$ and $\theta_{v2}$ -$\theta_{v1}$ may be 1–2.5 km and 0.5–3 K, respectively. The values of $\theta_{v2}$ and $V_c$ in (8) are obtained from the ERA reanalysis data nearest the location of each outer TCR observed in this study.

The calculations indicate that the percentage of the observed propagations that fit into the corresponding velocity ranges for the atmospheric cold pools is equal to 77.7% and 36.3% for the squall-line TCR group and non-squall-line TCR group, respectively. The contrasting percentage of the fit between the two TCR groups can be clearly seen from the statistical distribution of observed propagations

and theoretically predicted velocity ranges of cold pools for each of the TCR cases, as shown in Fig. 5c, d. The observed propagation velocities for the squall-line TCR group (small blue circles) correspond generally to the cold pool velocity predictions (red shading) (Fig. 5c), whereas most of the observed propagation velocities for the non-squall-line TCR group are distributed beyond the theoretical velocity ranges of atmospheric cold pools (Fig. 5d). Not only is it evident that there is a much higher percentage of agreement between the observed propagation velocities and theoretically calculated cold pool velocities for the squall-line TCR group compared to that of the non-squall-line TCR group, but the majority of the TCR cases for the squall-line TCR group (530 cases out of 682 cases) also tend to have a cold pool-like propagation velocity, both of which suggest the relevance of squall-line dynamics to the squall-line TCR group. In addition, it has long been

recognized that one of the representative characteristics for squall-line convective storms is their fast movement[74,76]. Compared to non-squall-line convective systems, one might see a more rapidly propagating nature for convective systems if they evolve with the operation of squall-line dynamics. Consistent with this expectation, the box and whisker plot of the observed propagation velocities, shown in Fig. 5b, indicates generally faster propagation characteristics for the squall-line TCR group than for the non-squall-line TCR group. The mean propagation velocities calculated from the squall-line and non-squall-line TCR groups are equal to 6.1 and 3.5 m s$^{-1}$, respectively. According to the Student's $t$ test, such a velocity difference between the two TCR groups is calculated to have statistical significance at 99% confidence level. The assessment here provides further evidence supporting the importance of squall-line dynamics for the squall-line TCR group.

Two important implications may emerge from the above results. First, as shown in Fig. 5, a considerable number of the observed outer TCRs (i.e., the non-squall-line TCR group) do not actually exhibit squall-line environmental and propagation characteristics. The formative scenario closely related to squall-line dynamics is expected to occur frequently but not exclusively. Formative mechanisms other than squall-line dynamics must have existed and deserve future exploration. Second, in addition to the squall-line environment, the appearance of cold pools is equally important and can be treated as one of the key preconditions needed to start the scenario in which squall-line dynamics leads to the formation of outer TCRs. The cold pool signatures have been commonly observed in the vicinity of outer TCRs during their developing or mature stage and are most likely contributed by either the evaporative cooling associated with TCR precipitation or by the downward transport of low equivalent potential temperature air aloft via convective downdrafts[9,13,28,30,55]. However, during the preformation stage of an outer TCR, its associated precipitation is inherently weak, with scattered and disorganized distributions and a very limited horizontal extent (Fig. 9a presented in the Methods section). The aforementioned cooling mechanisms driven by the nascent TCR itself should be generally ineffective, and there must be other preexisting precipitation (i.e., separating from the considered outer TCR), which could act as an initial, critical provider of cold pools to initiate the operation of squall-line dynamics and contribute to the genesis of outer TCRs. In this context, the initiation of outer TCRs is expected to preferentially take place adjacent to some certain precipitation areas of TCs where evaporative cooling and/or convective transport is potentially active.

To validate the speculation above, an effort is made to construct the spatial composite of precipitation frequency (>15 dBZ) with respect to the location of each outer TCR at three particular time periods, 1 h prior to the preformation time ($t_0$-2), the preformation time ($t_0$-1) and the formation time ($t_0$). The preformation time is defined as the earliest, unambiguous detection of TCRs by radar observations whereas the formation time refers to a time satisfying the formative criteria, as described in the Methods section. In this analysis, we choose the squall-line TCR cases with faster propagation velocities (i.e., greater than the mean cross-band propagation velocities of 6.1 m s$^{-1}$ calculated from the squall-line TCR group) and corresponding to the theoretical velocity range of atmospheric cold pools to maximize composite signals for the squall-line outer TCRs. Similarly, those non-squall-line TCR cases that propagate relatively slower (i.e., less than the mean cross-band propagation velocities of 3.5 m s$^{-1}$ calculated from the non-squall-line TCR group) and beyond the theoretical velocity range of atmospheric cold pools are chosen for the non-squall-line composite. As shown in the composite results, the position of the rainband is clearly marked by the highest precipitation frequency centered at ($x = 0$ km, $y = 0$ km) at $t_0$-1 and $t_0$ for both the squall-line TCRs and non-squall-line TCRs (Fig. 6). For the squall-line TCR composite, there is a concentrated area of preexisting precipitation on the rear flank of the rainband prior to its formation at $t_0$-2 (Fig. 6a). In

particular, the rainband tends to be initiated immediately adjacent to the outer boundary of the preexisting precipitation in the outer regions of TCs (Fig. 6b, c), consistent with our speculations above. For the non-squall-line TCR composite, there are generally scattered distributions of lower precipitation frequencies (18–30%) over a broad region in both front and rear sides of the rainband (Fig. 6d–f). These features suggest that the rainband formation occurs in a relatively precipitation-free area and is not frequently associated with prominent preexisting precipitation. The contrasting statistical distribution of outer precipitation between the squall-line and non-squall-line TCR composites further supports that the preexistence of suitable precipitation in the outer region of TCs would be critical to provide important sources of cold air to initiate the operation of squall-line dynamics for the squall-line outer TCRs. A schematic diagram is also prepared, as shown in Fig. 7, to help illustrate the two distinct types of outer precipitation and their associated formative scenarios of squall-line and non-squall-line outer TCRs.

In addition, given that the lack of consistency in the propagation characteristics between the observed outer TCRs and VRWs/GWs, as elaborated earlier, it seems unlikely that the VRWs/GWs play a direct role in triggering the outer TCRs. Because the generation of some convection/precipitation within TCs could be related to these wave activities, we cannot entirely rule out the wave-induced precipitation as a potential contributor that would favor the formation of cold pools and then initiate the operation of squall-line dynamics. To evaluate this likelihood and its underlying processes is beyond the scope of the study, but the potential, indirect roles of wave activities in facilitating the formation of outer TCRs deserve future investigation through detailed case studies using high-resolution observations and numerical simulations.

## Discussion

The origin of outer TCRs is a long-standing, unresolved topic in the TC research community. To explore this issue, this study performs analyses on a total of 1029 outer TCRs identified during their formative stage by long-term radar observations collected near Taiwan from 2002 to 2019. The observations presented in this article provide a robust foundation of our knowledge regarding the natural diversity of the outer TCR origin.

One of the striking findings is that current theories of outer TCRs cannot appropriately explain the observed TCR characteristics and statistics. A dominant portion of the identified TCRs (97%) tend to be initiated locally in the outer TC region. This outer-origin dominance is in distinct contrast to numerous theoretical modeling studies of outer TCRs that propose inner-origin scenarios. Moreover, the formative fractions of the identified outer TCRs within and outside the storm radius of TCs (R34) are dramatically different, equal to 29% and 71%, respectively, indicating that outer TCR formation occurs mostly in regions beyond the operational TC alert area. A dominant majority of the identified TCRs possess radial propagation velocities within the range of ±10 m s$^{-1}$, with evidence for both outward and inward propagation. Compared to the radial propagation, the tangential propagation of outer TCRs is generally much faster and predominantly cyclonic, which is consistent with the influence of TC vortex circulations. Large discrepancies between these observed propagation characteristics and theoretically predicted propagation velocities of both GWs and VRWs are found, suggesting that wave disturbances do not have a direct impact on the origin of the observed outer TCRs. This result contrasts sharply with inner TCRs that have been recognized to be closely related to wave activities triggered near the eyewall.

Analyses from this study provide supportive evidence that squall-line dynamics may play an important role in contributing to the formation of outer TCRs. The observed outer TCRs appear to have a statistical preference for both a squall-line environment and cold pool propagation behavior. In particular, the initiation of faster, squall-line-

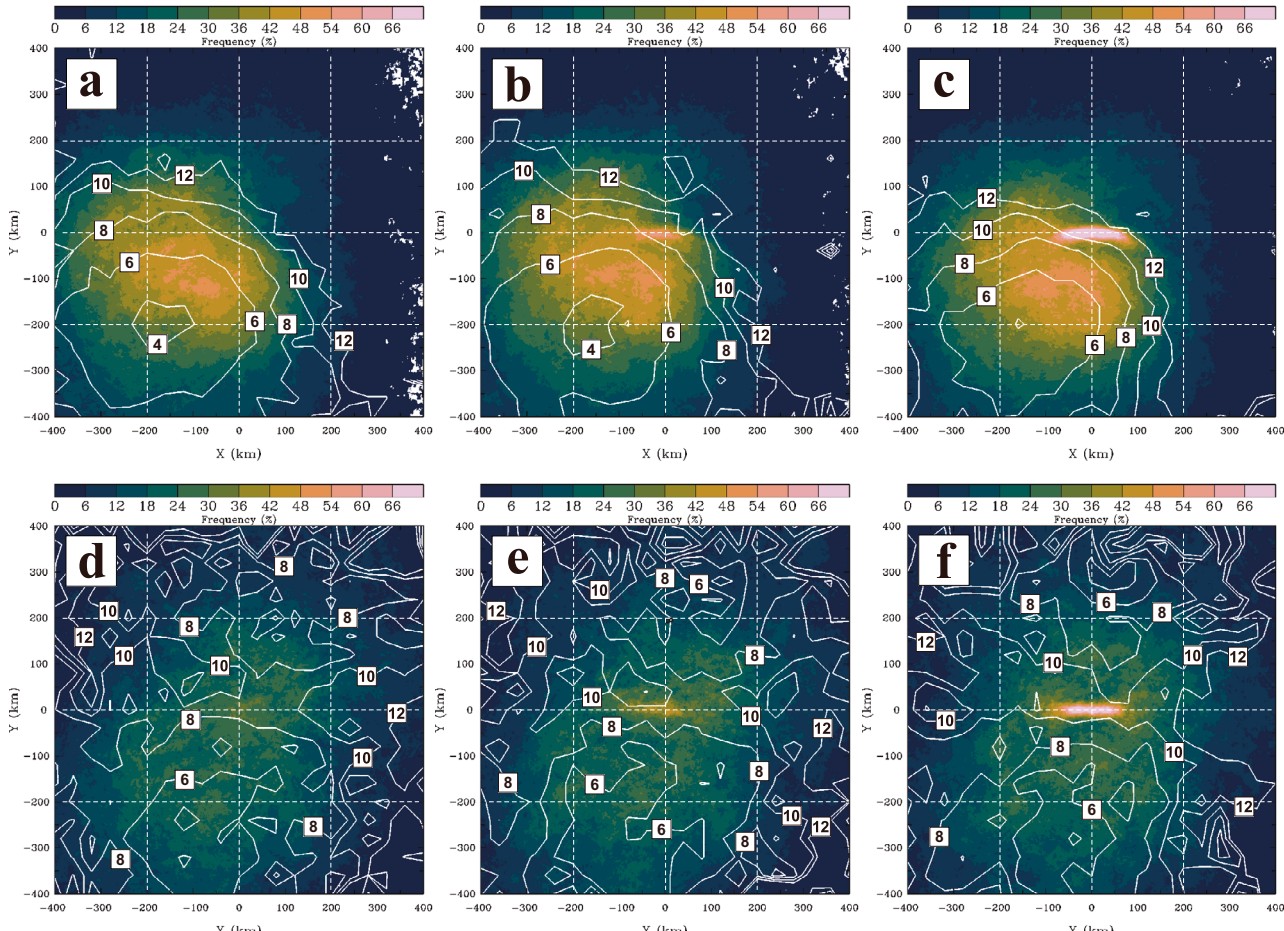

**Fig. 6 | Composite precipitation distributions of squall-line and non-squall-line tropical cyclone rainbands (TCRs) during and before their formation.** The spatial composite of precipitation frequency (>15 dBZ, color shading) and normalized radial distance by the radius of maximum wind (RMW, contour) with respect to the location of each outer TCR ($x = 0$ km, $y = 0$ km) for 303 squall-line TCR cases at (**a**) 1 h prior to the preformation time ($t_0$-2), (**b**) the preformation time ($t_0$-1) and (**c**) the formation time ($t_0$). **d**–**f** as in (**a**)–(**c**), but showing the composite results for 140 non-squall-line TCR cases. For the composites at $t_0$-1 and $t_0$-2 in (**a**), (**b**), (**d**) and (**e**), the location of each TCR case is determined by the backward extrapolation from the TCR location at the formation time (i.e., $t_0$) based on the calculated propagation velocities of TCRs.

like outer TCRs preferentially take place immediately adjacent to the outer boundary of prominent precipitation in the outer regions of TCs, and this preexisting outer precipitation may act as an initial, critical provider of cold pools to activate the operation of squall-line dynamics (Fig. 7a). Examining the causes of outer precipitation and clarifying its roles in contributing to the occurrence of cold pools during the preformation stage of rainbands would thus be a necessary task that would help improve our understanding of the initiation of outer TCRs. On the other hand, the formative scenario closely related to squall-line dynamics is expected to occur frequently but not exclusively because a considerable number of the observed outer TCRs do not actually exhibit squall-line environmental and propagation characteristics. These non-squall-line outer TCRs propagate relatively slower and tend to be initiated in the precipitation-free or widely-scattered, weak precipitation area (Fig. 7b). Formative mechanisms other than squall-line dynamics must also exist and deserve future exploration.

Despite the different evolving stages of outer TCRs, the cold pool signatures associated with outer TCRs have been reported for TCs developing over different oceanic basins, such as the northwestern Pacific Ocean[28,29] and Atlantic Ocean[53,55]. In addition, outer TC circulations observed over different basins seem to have similar characteristics of low-level winds with the presence of some band-normal vertical shear[29,52,54]. It is thus reasonable to expect that what we learn from the study would be generally applicable to outer TCRs

developing over other basins. However, a comprehensive dataset of outer TCRs collected from different geographical locations during their formative stage will be still required to clarify whether any possible basin-dependent characteristics of the TCR origin may exist.

Although this study provides important findings and implications, our understanding on the origin of outer TCRs is clearly incomplete. Future efforts and observational programs should pay special attention to the outer-origin processes and the identification of possible cold pool sources by collecting more complete observations of three-dimensional kinematics, thermodynamics and precipitation in the outer environment of TCs during a very early stage of rainband development. Making these datasets available, together with high-resolution numerical simulations, is critical for developing a theoretical model of how squall-line dynamics operates to form outer TCRs and to seek other unidentified processes responsible for the initiation of outer TCRs.

## Methods
### Data
The datasets used to identify TCRs in this study are provided by four S-band (10 cm) ground-based Doppler radars operated by the CWB at Wu-Fen-San (WFS), Chi-Gu (CG), Ken-Ting (KT) and Hua-Lien (HL), which are located in different coastal regions of Taiwan. The basic characteristics of these Doppler radars are described in Yu et al.[29], and

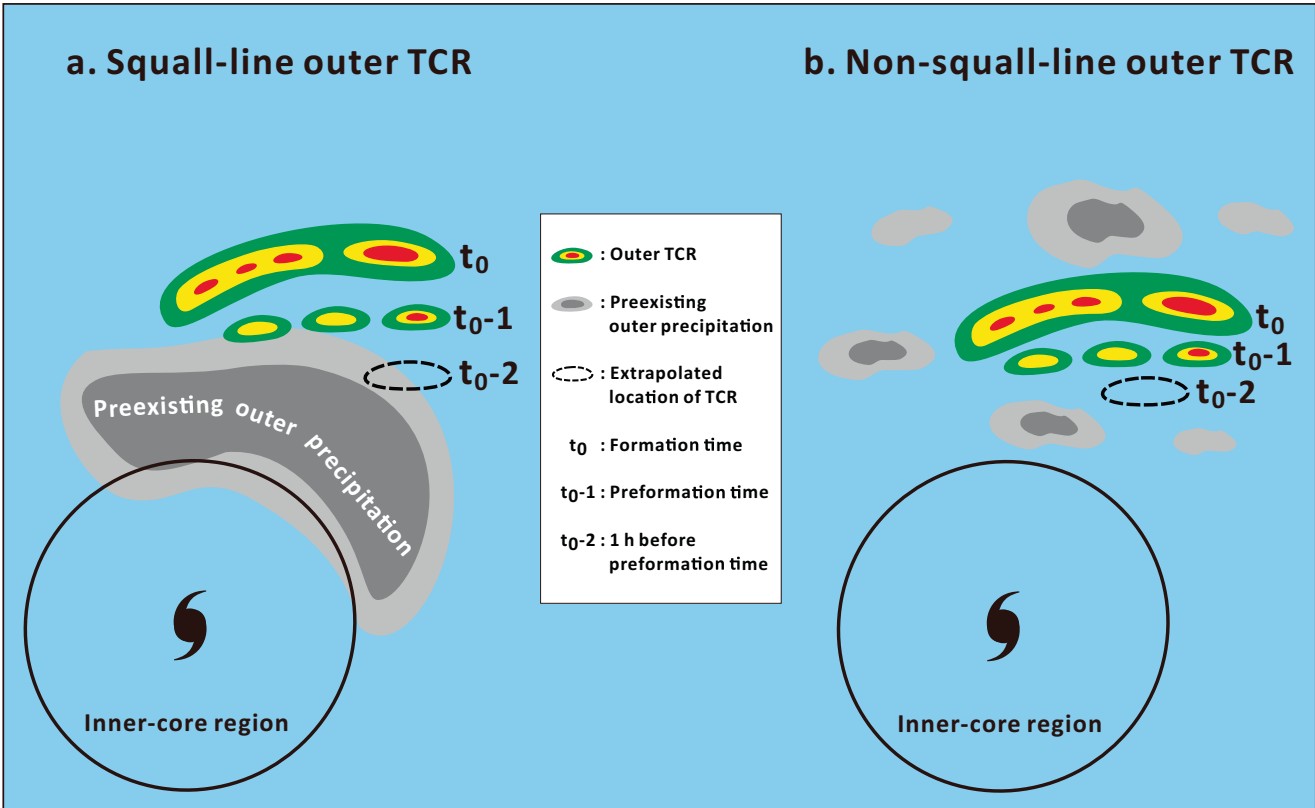

**Fig. 7 | Two distinct types of outer precipitation associated with squall-line and non-squall-line outer tropical cyclone rainbands (TCRs) during their formation.** Schematic diagram illustrating the formative scenario for the squall-line outer TCR (**a**) and the non-squall-line outer TCR (**b**), and their embedded precipitation characteristics are particularly highlighted. The solid circle denotes the inner-core boundary of tropical cyclones.

the locations of the four radar sites are shown in Fig. 8. These coastal radars provide volumetric scans of reflectivity, with a temporal interval ranging from 6 to 10 min between each volume, scanning elevations from 0.4° to 19.5° and a maximum observational range of ~460 km. The data coverage from these radars encompasses broad regions, extending from the island of Taiwan to approximately 300–400 km offshore (cf. Fig. 8). In addition to the radar measurements described above, the fifth-generation European Centre for Medium-range Weather Forecasts (ECMWF) reanalysis data (ERA5[77]) are also used to determine various environmental factors associated with the studied TCRs. Compared to other reanalysis data, the ERA5 reanalysis data have higher spatial and temporal resolutions at 0.25° × 0.25° and 1 h, respectively, thus providing a more realistic representation of TCs and their environment[78–80]. For convenience of relevant calculations and analyses, the ERA5 reanalysis data valid at different pressure levels are interpolated linearly to obtain vertical grid spacing at 0.1 km. The TC information, including center locations, RMW, and other storm characteristics, analyzed in this study comes from the best track data provided by the Joint Typhoon Warning Center (JTWC).

Although ERA5 data have been widely used to study relevant scientific issues for TCs, whether they can provide adequate representation of wind information nearby the studied outer TCRs is still somewhat uncertain. To fully address this issue is beyond the scope of this study, but the possible impact of this uncertainty on the calculated propagation velocity of VRWs presented in the Results section may be evaluated by comparing the ERA5 winds with other independent wind measurements nearby the rainband's location. In making such comparisons, surface observations from eight offshore island stations [Pengjiayu (PJY), Yonaguni (YO), Hateruma (HTM), Iriomote (IM), Ishigaki (IG), Green island (GI), Penghu (PH) and Dongji (DJ)] near Taiwan (locations in Fig. 8) within a distance of 35 km from the formative

location of outer TCRs are first collected. The reason why the threshold distance of 35 km is used herein is because it is approximately comparable to the grid spacing of ERA5. The island observations represent the ground truth measurements and are available from a total of 70 outer TCR cases. The near-surface winds at the 10-m (MSL) level from the ERA5 gridded data closest to the island stations for these TCR cases are then chosen to compare with station-measured winds. Firstly, we focus on the comparison of the tangential wind [i.e., $V$ in (5), (6) and (7)] that is the ERA5-provided kinematic quantity involved with the calculation of VRWs.

A good correlation between the ERA5 data and offshore island observations is found, and the correlation coefficient, average error and root mean square error (RMSE) are calculated to be 0.74, 2.9 m s⁻¹ and 4.5 m s⁻¹, respectively. If we assume the calculated RMSE as potential errors for the tangential winds from ERA5, potential velocity errors in radial and tangential directions are estimated to be 0.2 and 4.4 m s⁻¹, respectively, based on (2)–(7). The potential errors in the calculated propagation velocity for VRWs may be considered minor because they are unlikely to alter the statistical distributions of the scatterplots shown in Fig. 4b, c significantly.

Using the similar method of evaluating the uncertainty associated with ERA5-provided tangential winds in the rainband environment, the potential error of $V_c$ (i.e., the band-normal wind component) from ERA5 is calculated be 2.9 m s⁻¹, yielding a velocity error of 1.7 m s⁻¹ in the estimate of theoretical cold pool velocity ($V_{cp}$) based on (8).

The error analyses above support the reasonable representation of the ERA5 reanalysis data for the rainband environment, although the accuracy of the ERA data to represent TC circulations and their ambient wind conditions is possible to vary case by case. This uncertainty may have greater chances to become a potentially serious problem if scientific approach is placed on the understanding of individual

TC/TCR events. However, because the present study focuses on the statistical aspects of outer TCR formation using long-term reanalysis data for a very large set of TCR cases, the potential impact of this data uncertainty may be appropriately mitigated.

## Rainband identification

To seek TCRs, a very large set of radar reflectivity images and data from the lowest plan position indicator (PPI) scans (elevation ~0.4°) for TCs approaching Taiwan is initially prepared. The dataset comes from the four radars and contains observations of 95 TCs for which the CWB issued typhoon warnings during 2002–2019. All of the reflectivity images produced are first screened visually to determine potential candidates of TCRs whose radar echoes can be tracked backward from their mature to formative stages. The primary purpose of this subjective screening is to retain TCR cases that could be captured successfully by radars during their initiation time. Reflectivity data from these tentatively chosen TCR cases are then used to calculate basic rainband information, such as location, length and width, for each of the TCRs. The specific location of rainbands at a given time is determined by averaging the azimuthal angles and radial distances with respect to the TC center over the rainband area with reflectivities ≥30 dBZ. Given the convective nature of outer TCRs, the 30 dBZ threshold not only represents the minimum requirement of precipitation intensity used to identify outer TCRs as described below but also appropriately increases the weighting of stronger and/or convective precipitation on the location determination. The length of rainbands is determined by the distance between two ends of the long axis of the rainband area with reflectivities ≥20 dBZ. The reason why a lower reflectivity threshold of 20 dBZ is used here is because it encompasses a more continuous, larger spatial extent of radar echoes and provides a better representation of the entire rainband entity. Similarly, the width of rainbands is approximated by a mean distance in the band-normal direction along the rainband entity. With these calculated rainband characteristics above, it is possible to identify an outer TCR based on objective, quantitative criteria. The criteria used to identify a TCR are similar to those used by previous TCR studies[9,29] but more specific criteria are used herein to define an outer TCR and determine its initiation. First, a precipitation feature with a maximum reflectivity of at least 30 dBZ, a minimum length of 100 km and a large length-to-width ratio (>5) must be satisfied. Second, the precipitation signature described above must persist for 1 h or longer. These two criteria can assure that a prominent and persistent rainband has been identified, and that a relatively transient, non-banded or loosely organized precipitation feature could be effectively avoided. Third, a rainband is classified as an outer TCR as long as its precipitation occurs in or propagates into regions beyond a radial distance of 3 times the RMW for a certain time period during rainband formation and development. This criterion allows us to include both possible formative scenarios: one is that the rainband begins development within the inner TC core and then propagates outward over time into the outer region, and the other is that the rainband is initiated locally in the outer region. Fourth, the formation time for an outer TCR is defined as when the rainband initially satisfies the first criterion above.

It is noteworthy that Taiwan rainbands[81,82] that sometimes occur along the coastal zone of eastern Taiwan in the outer region of TCs due to orographic effects may contaminate the TCR dataset. One of the most common features of these topographically generated rainbands is that they are oriented approximately parallel to the coast without obvious curvature. In addition, these rainbands tend to be locked with coastal mountains and do not propagate rapidly with cyclonic circulations of TCs. These characteristics contrast sharply with evolving and propagation aspects of TCRs. Hence, if rainbands are observed to form locally within the coastal zone (from coastline to ~25–100 km offshore)

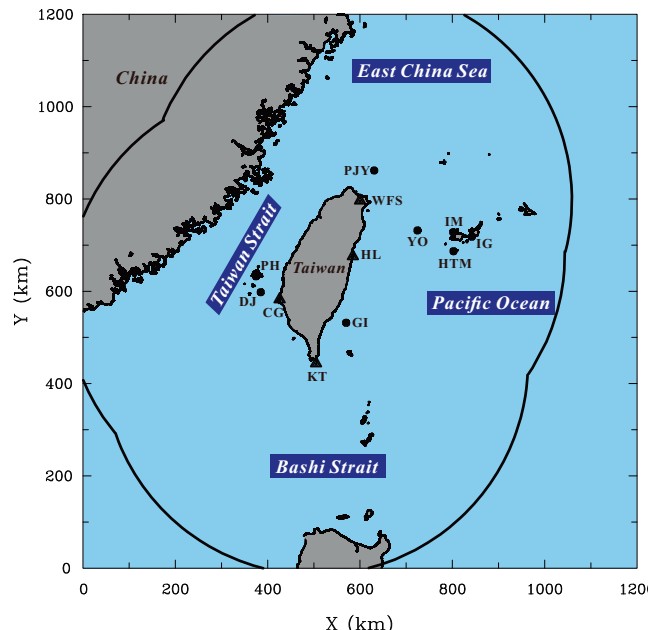

**Fig. 8 | Radar dataset and surface observations used in this study.** The locations of the coastal Doppler radars at Wu-Fen-San (WFS), Hua-Lien (HL), Chi-Gu (CG) and Ken-Ting (KT) are denoted by triangles. The locations of offshore island observations near Taiwan at Pengjiayu (PJY), Yonaguni (YO), Hateruma (HTM), Iriomote (IM), Ishigaki (IG), Green island (GI), Penghu (PH) and Dongji (DJ) are marked by solid circles. The outer curve denotes the approximate data coverage for these radars, extending from the island of Taiwan to 300–400 km offshore.

and meet the above signatures, they are assumed to be topographically generated rainbands and are thus removed during TCR identification.

A sample plot of the horizontal reflectivity patterns of an evolving outer TCR associated with Typhoon Saola on 31 July 2012 is shown in Fig. 9 to illustrate the typical scenario of rainband formation and development. The observed TCR started with well-separated, relatively weak reflectivity cells ~150 km northwest of the TC center at 1830 UTC (Fig. 9a). This was the earliest, unambiguous detection of the rainband by radar observations, so this period was referred to as the preformation time. A visually banded feature of radar reflectivity was initially evident at 1930 UTC (Fig. 9b), which was defined as the formative time for this particular rainband since it met the criteria described above. Note that the outer TCRs identified in this study usually did not take long to evolve from the preformation to formative stage, similar to the presented rainband (~1 h) (Fig. 9a, b). The mean time required for this transition among all identified TCRs is calculated to be 0.9 h. With time, the rainband intensified persistently, and a number of new reflectivity cells that developed in the rainband vicinity were observed, which formed an elongated, arc-shaped zone of heavy precipitation (Fig. 9c). The rainband reached the mature stage with maximum length (~400 km) and intense reflectivity (>45 dBZ) at ~2130 UTC (Fig. 9d).

With the comprehensive examination of the available radar measurements collected from 2002 to 2019, a total of 1029 outer TCRs satisfying the above criteria are identified from 95 TC events. These identified TCR cases are all successfully captured and tracked by radar observations over a reasonable time period from their formative to developing/mature stage, which is expected to provide important and unique clues for the possible origin of outer TCRs. A complete list of 1029 identified TCR cases is provided in Supplementary Data 1. All of the outer TCRs identified by radar observations in this study have also been checked if they are primarily influenced by the cyclonically rotational circulations of TCs and are

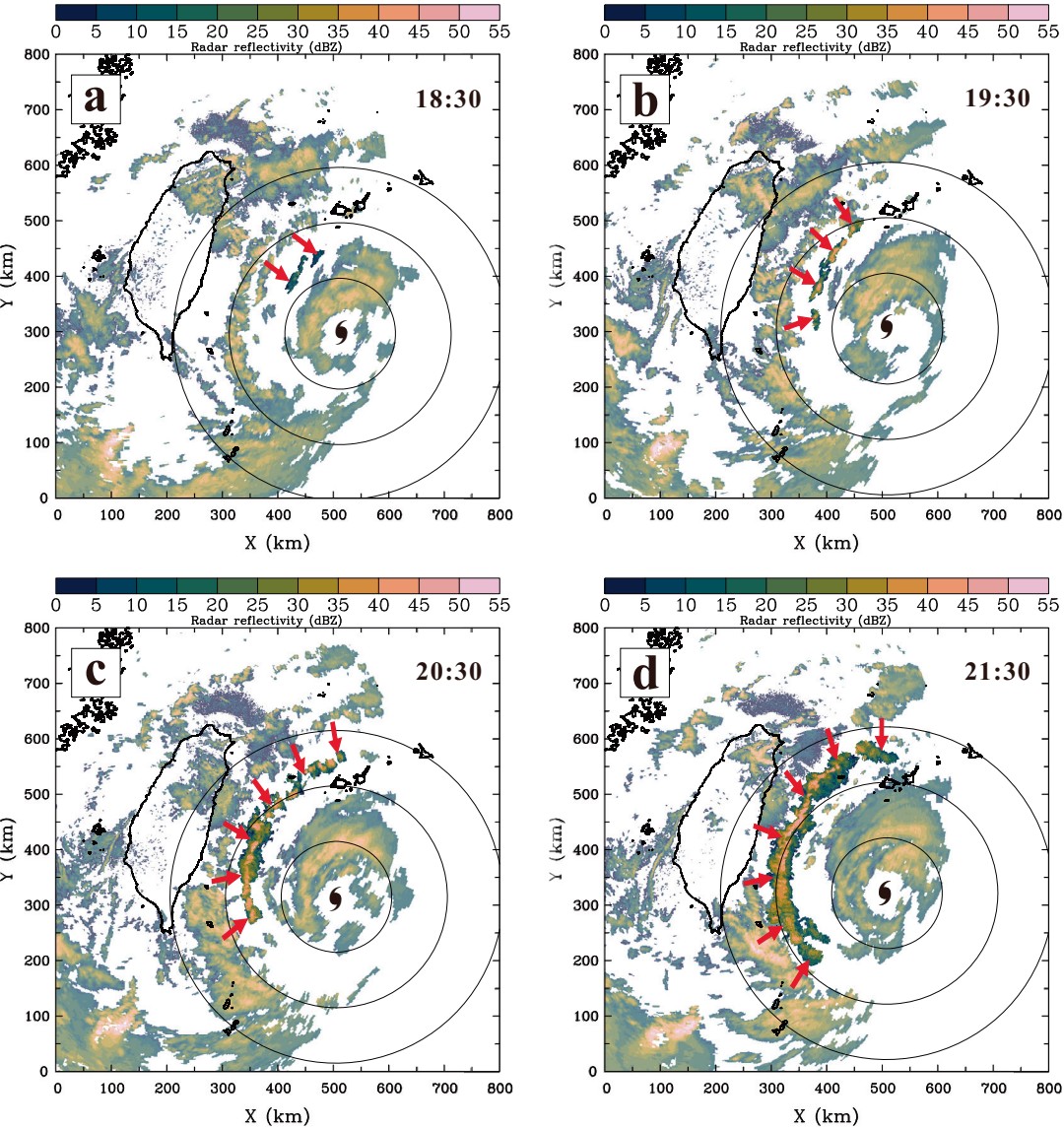

**Fig. 9 | Precipitation evolution of an outer tropical cyclone rainband (TCR) observed during its preformation to mature stage.** Low-level composite radar reflectivity (dBZ, color shading) associated with Typhoon Saola at (**a**) 1830, (**b**) 1930, (**c**) 2030 and (**d**) 2130 UTC on 31 July 2012. In each panel, the red arrows mark the location of the studied outer TCR. The rainband is also highlighted by reducing the transparency of the rest of the radar reflectivities. Gray range rings with respect to the tropical cyclone center are indicated every 100 km.

not associated with any other synoptic weather systems (e.g., fronts or cyclones). This checking is practically done by looking at both weather analysis maps provided from CWB of Taiwan and atmospheric circulations nearby TCs seen from the ERA5 reanalysis data, which can ensure that the identified outer TCRs belong to precipitation features exclusively induced by TC forcings.

The radial and tangential propagation velocities of each rainband are then obtained by calculating the difference in the rainband's location between the formative and mature period of the identified TCRs divided by the corresponding time interval. The reliability of the propagation estimate above relies mainly on the accuracy of the TC center positions. Note that if such a position error remains unchanged with time, there is no effect on the propagation estimate. Evaluation of the uncertainty of the TC centers provided by JTWC may be practically performed for some particular periods when the center vicinity of the TCs is well captured by radar observations so that the TC centers can be determined by calculating the geometric center of the weak echo regions surrounding the TC eye. Based on a group of outer TCRs (174 rainband cases from 39 TCs) with available radar-determined TC centers, the mean difference in the propagation velocities of the observed outer TCRs calculated from JTWC best track data and radar-determined center information are calculated to be 0.7 m s⁻¹ in both the radial and tangential directions. Given no perfect observation is available to estimate the TC motions, the differences calculated herein do not actually represent the errors, and they just imply a possible uncertainty for the estimation of TC motions based on independent datasets and methods.

## Data availability
Radar observations are provided by the Taiwan Central Weather Bureau and the radar data used for analysis in this study are available from the corresponding author upon request. The fifth-generation European Centre for Medium-range Weather Forecasts (ECMWF) reanalysis data are available from the research data archive (https://doi.org/10.24381/cds.bd0915c6). The TC best track

archive of the Joint Typhoon Warning Center is available at https://www.metoc.navy.mil/jtwc/jtwc.html?best-tracks. Source data are provided with this paper.

## Code availability

The data processing and analysis codes are available from the corresponding author upon request.

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

## Acknowledgements

The ground-based Doppler radar data used in this study were provided by the Taiwan Central Weather Bureau (Central Weather Administration since 15 September 2023). The TC information, including center locations, RMW and other storm characteristics, was provided by the Joint Typhoon Warning Center. This study was supported by the National Science and Technology Council of Taiwan under Research Grants MOST 111-2111-M-002-017 and NSTC 112-2111-M-002-017.

## Author contributions

C.-K.Y. conceived and led the study, interpreted the results and wrote the manuscript, C.-Y.L. helped with data collection, performed data analysis and prepared figure illustrations, and C.-H.P. helped with data collection and part of initial data analysis.

## Competing interests

The authors declare no competing interests.
