## [Peer Review File · Nature Communications]

Origin of outer tropical cyclone rainbandsREVIEWER COMMENTS

Reviewer #1 (Remarks to the Author):

Summary:

The authors present a very nice radar climatology of outer-core tropical cyclone rainbands from 95 tropical cyclones near Taiwan from 2002-2019. Tropical cyclone rainbands are an important hazard and therefore this extensive climatology of their development and evolution is of value to the community. The authors use this radar climatology to document that outer-core tropical cyclone rainbands, defined as existing at least three times the radius of maximum winds from the storm center, predominantly develop within the outer-core region and do not commonly propagate outward from the inner-core region. The authors also use ERA-5 reanalysis data to suggest that the outer-core rainbands often propagate in a manner consistent with squall line dynamics and not consistent with vortex Rossby waves or gravity waves. Despite some potential limitations from using the relatively coarse ERA-5 reanalysis data I believe this analysis is appropriate.

My concerns are primarily that (1) I believe the authors misuse the word 'origin' by not providing a convincing description of what initiates the outer-core rainbands, (2) a lack of discussion on the diurnal cycle which is well established as influence tropical cyclone convection, and (3) a lack of discussion on vertical shear. Regarding (1), while I believe the authors adequately show that the convection associated with the outer-core rainbands develops within the outer-core region of the tropical cyclone, they do not provide a clear mechanism for their initiation. Instead, I believe the authors incorrectly use squall-line dynamics as an explanation, which I believe to instead be an explanation for their maintenance once convection is ongoing, but not their formation. With these concerns addressed, I believe the study can provide a useful climatology of tropical cyclone rainbands to the community.

Major Comments:

1) I am concerned by the title of the paper and the use of the word 'origin' throughout the article. In my opinion the authors do not clearly show the origin of outer tropical cyclone rainbands. Rather I believe the authors demonstrate that the outer rainbands are often maintained through cold pool dynamics, not initiated by them. I fail to see the authors clearly demonstrate what initiates the outer rainbands. As a result, I believe the authors overstate their results such as in L44-46, L245-247, L503-504, L508-510. I also am not entirely convinced that the authors can rule out the role of gravity waves, vortex Rossby

waves, or something else in triggering the initiation of the outer rainband convection before the cold pool dynamics help to support their maintenance thereafter (L497-502).

2) I understand that the authors used available radar data from Taiwan, but a natural question from readers will be how applicable are these results to tropical cyclones in other basins?

3) One aspect missing from the introduction and the analysis is the diurnal cycle, which is well known to influence tropical cyclone convection beyond the inner-core region (e.g., Dunion et al. 2014) The authors should consider if there is a diurnal cycle to outer-rainband formation and if the diurnal cycle may be an important mechanism to their formation.

Dunion, J. P., C. D. Thorncroft, and C. S. Velden, 2014: The Tropical Cyclone Diurnal Cycle of Mature Hurricanes. *Mon. Wea. Rev.*, 142, 3900–3919, <https://doi.org/10.1175/MWR-D-13-00191.1>.

4) A second aspect that appears to be missing from the analysis is a discussion on vertical wind shear, which is well known to influence tropical cyclone convection (e.g., Corbosiero and Molinari, 2002). I recommend adding an additional panel (Figure 3) that shows the formation locations relative to the deep-layer shear in an RMW relative coordinate. The criteria mentioned for the classification of an outer tropical cyclone rainband required the rainband forming or propagating into the region beyond 3 times the RMW. Showing the formation locations relative to the RMW and shear may yield additional insights.

Corbosiero, K. L., and J. Molinari, 2002: The Effects of Vertical Wind Shear on the Distribution of Convection in Tropical Cyclones. *Mon. Wea. Rev.*, 130, 2110–2123, [https://doi.org/10.1175/1520-0493\(2002\)130<2110:TEOVWS>2.0.CO;2](https://doi.org/10.1175/1520-0493(2002)130<2110:TEOVWS>2.0.CO;2).

5) I think the use of ERA5 for the TC presents potential for criticism. This may be unavoidable, but the authors should add some discussion from using .25° data to represent the TC conditions and the associated gravity wave and vortex Rossby wave calculations within their analysis (L370-373, L432-433). As an example, can we trust filamentation time estimated from relatively coarse ERA5 data?

Minor Comments:

L41: I recommend adding “radar” before “observations”.

L53: “storm” should be changed to “storms”.

L81: I think “inner-core” should be “outer-core”.

L94: “would probably” can be changed to “could” and “but” can be removed.

L100: “actually” can be removed.

L102: “more” can be removed.

L127: “evidence” can be removed.

L184-186: It should be mentioned how the orographically generated rainbands were removed from the analysis.

L233: Do the authors mean this is a result of the Taiwan radars more frequently sampling the front quadrants as tropical cyclones approach Taiwan?

L264-268: The size of a tropical cyclone is often considered to be larger than the radius of tropical storm force winds. As mentioned by the authors other metrics are also commonly used including the outermost closed isobar and the radial extent of weaker wind speed thresholds, such as 10 or 12 m/s. The authors should be careful in defining tropical cyclone size as the radial extent of tropical storm force winds (34 kts). See also L490.

L283: I recommend adding “near Taiwan” following “outer TCRs”.

L284: “crossing angles” needs to be defined somewhere.

L357: “actually” can be removed.

L386: “circumstances to activate the operation” can be removed.

L462,510: I think the authors mean for “predominantly” to be “exclusively”.

L465,504,519: The authors show that squall-line dynamics are often important for the maintenance of the rainbands, not the “formation”.

Reviewer #2 (Remarks to the Author):

Review of "Origin of outer tropical cyclone rainbands"

Synopsis:

This is an important study that utilizes a radar dataset to investigate the formation location and movement of outer tropical cyclone (TC) rainbands, as well as potential formation mechanisms. As discussed in the introduction, there are competing theories on the formation and origin of outer TC rainbands, with numerous (mostly modeling) studies indicating that these rainbands are generated as a result of TC inner core dynamic. In the current study, the authors provide convincing evidence that most outer TC rainbands form outside the inner core, and in some cases outside the radius of gale force winds. Additionally, the rainbands propagate at radial and tangential velocities that are inconsistent with inner origin theories. Instead, the authors show that a majority of outer TC rainbands propagate in a manner consistent with squall line (and cold pool) dynamics. Therefore, I believe that this study is an important contribution to the research literature. However, I have major concerns associated with data and methods, which are outlined below. Specifically, use of ERA5 data in certain parts of the study is not justified and not likely to be sound. As a result, I recommend major revisions for this manuscript.

Major comments:

1. Use of ERA-5 data: On lines 158-163, the authors provide some basic information about the ERA-5 dataset and justify its use for it "more realistic representation of TCs and their environment." It is generally true that the ERA-5 dataset provides more realistic data about the TC environment as well as some other TC-related metrics such as TC size. However, in this study, the authors have used ERA-5 data to capture information about lower-level (boundary layer) winds that are unlikely to be accurate due to the low resolution of the associated ERA5 model and the need for these winds to be calculated through parameterization. Could the authors find another dataset that would provide more accurate low-level wind info? Later in the results, the authors use ERA5 data to assess whether cold pool dynamical (and thermodynamical) processes are present, and these mesoscale processes are very unlikely to be captured by the low-resolution (0.25°x0.25°) ERA5. I have major concerns about using ERA5 to assess mesoscale squall-line processes.
2. Please provide more information about your "subjective screening" of the radar data to locate outer TC rainbands.
 - a. As described in the text, the first set of criteria include measurements of the rainband and other quantitative criteria, which suggest that it could have been applied algorithmically. Or was this part applied manually too? If manual, please provide justification for doing so. Manual/subjective application of the length-to-wind-ratio and minimum length seems less than ideal.
 - b. Do your data include the principal rainband? or have you excluded it using your subjective analysis?

c. Lines 184-186: Please elaborate on how you identified orographically generated rainbands.

d. Lines 236-237: “Nevertheless, considerable TCRs are observed to form in far outer TC regions beyond the radius of 350 km.” How do you identify whether these rainbands are associated with the TC rather than some other forcing? Please provide more details in the methods to describe how you decipher whether the rainband is associated with the TC. Rainbands at distance ~1000 km from center seem like they could be forming due to some other mechanism.

Minor comments:

Line 37: I don’t agree on the outer rainbands being the “most persistent.” The principal rainband is more persistent. It might be better to focus on the hazardous weather part of this statement (or the extreme rain rates observed in these rainbands). For example, you mention in the article that these rainbands can exist outside the TC gale force wind radius.

Line 45: squall line dynamics “are” a common mechanism.

Line 46: I don’t think the word “road map” is the right choice because you don’t specifically outline research directions and how those should be navigated.

Line 54: There is not a clear connection between sea level rise and loss of life (it’s leading to enhanced storm surges that lead to loss of life). Suggest rewording.

Line 64-68: I disagree with characterization of inner vs. outer rainbands as transient (inner) and persistent (outer). Inner rainbands are usually dynamically driven and therefore persist (e.g., eyewall, principal rainband) whereas outer, convective rainbands are convective (thermo-driven) and less persistent in terms of their location and duration. Of course, some of the inner rainbands within the stationary band complex are more transient (such as secondary rainbands), but the eyewall and principal rainband in a major hurricane may be the most persistent and defining features of these storms. Some of the articles you cite (e.g., Willoughby et al. 1984) point out the persistent, stationary nature of the principal rainband. Convective storms can persist for hours but not over multiple days as the eyewall and principal rainband do.

Lines 68-70: This sentence is about the impacts on human and natural communities, but the references (28,29,31,32) are mostly about the structure of outer TC rainbands. The only cited article that seems relevant here is the one about meteotsunamis. I suggest finding studies that focus on rainband/flooding impacts instead.

Line 71: I think you might want to cite more or different studies here, at the very least Part II by Moon and Nolan (2015), which I believe is the more relevant study about the formation and propagation of inner rainbands.

Moon, Y., and D. S. Nolan, 2014: Spiral Rainbands in a Numerical Simulation of Hurricane Bill (2009). Part II: Propagation of Inner Rainbands. *J. Atmos. Sci.*, 72, 191–215, <https://doi.org/10.1175/JAS-D-14-0056.1>.

Line 73: There are a few more recent studies about observations of inner rainbands and the VRW theory. Suggest citing some here, primarily:

Guimond, S. R., P. D. Reasor, G. M. Heymsfield, and M. M. McLinden, 2020: The Dynamics of Vortex Rossby Waves and Secondary Eyewall Development in Hurricane Matthew (2016): New Insights from Radar Measurements. *Journal of the Atmospheric Sciences*, 77, 2349–2374, <https://doi.org/10.1175/JAS-D-19-0284.1>.

Line 75: delete extra word “be”

Line 78: “Despite” doesn’t seem to be the right word choice here. Maybe “Due to”? Think about rewording this sentence on lines 78-82 to better reflect the meaning you’re conveying.

Lines 93-95: I don’t follow how the principal rainband is related to outer or distant rainband formation. You say in the text “outer-origin scenarios would probably be related” to this rainband, but you don’t explain how. Could you elaborate? Or maybe there is no underlying theory and this sentence should be reworded or removed.

Lines 97-99: “Previous PB studies rely dominantly on the analyses of airborne flight-level and radar data collected over a relatively short period of time during the mature or late stage of oceanic TCRs.” There are multiple modeling and idealized studies of the principal rainband by Michael Riemer and colleagues. Please revise.

Riemer, M., 2016: Meso- β -scale environment for the stationary band complex of vertically sheared tropical cyclones. *Q.J.R. Meteorol. Soc.*, 142, 2442–2451, <https://doi.org/10.1002/qj.2837>.

Lines 101-102: “To the best of our knowledge, Yu et al. (2019) is the only observational study of TCRs to provide a more complete picture of the outer-origin processes.” This may be true. However, there are multiple studies that observe the cold pools associated with these rainbands and their squall-line like structure. I think you should also attribute the idea to these other studies, rather than citing them below as an afterthought (since the Yu et al. 2019 study came later and built on these former studies).

Eastin, M. D., T. L. Gardner, M. C. Link, and K. C. Smith, 2012: Surface cold pools in the outer rainbands of Tropical Storm Hanna (2008) near landfall. *Mon. Wea. Rev.*, 140, 471–491, doi:10.1175/MWR-D-11-00099.1.

Moon, Y., and D. S. Nolan, 2014: Spiral Rainbands in a Numerical Simulation of Hurricane Bill (2009). Part I: Structures and Comparisons to Observations. *J. Atmos. Sci.*, 72, 164–190, <https://doi.org/10.1175/JAS-D-14-0058.1>

Yu, C.-K. & Tsai, C.-L. Structural and surface features of arc-shaped radar echoes along an outer tropical cyclone rainband. *J. Atmos. Sci.* 70, 56–72 (2013).

Line 139: Can you provide a citation or website for these statistics about the number of landfalling TCs around Taiwan from the Taiwan CWB?

Line 143-145: The observations cannot “aim.” Reword this sentence.

Line 165: Please provide a website or reference for the JTWC best track data. It can be provided here in the text, but you may want to actually provide this information in the data availability statement.

Line 167-169: “The dataset comes from the four radars and contains observations of 95 TCs during 2002–2019, which the CWB issued typhoon warnings.” Here, you’re using “which” in a relative clause, but I think it needs to be used in a preposition: “The dataset comes from the four radars and contains observations of 95 TCs for which the CWB issued typhoon warnings (during 2002–2019).”

Lines 213-222: It's good to include these limitations. Could you provide even more information, such as the distribution of errors, not just the average value. It would be nice to see a table to histogram that summarizes this information more fully, since the average might obscure other important information about the full distribution, including very large error values due to a non-normal distribution.

Lines 232-233: “This weakly asymmetric characteristic may not be statistically robust and would be due in part to the relative configurations between the generally westward-moving TCs in the northwestern Pacific Ocean and the Taiwan area.” Elaborate and support this statement better. I don't understand why these would not be statistically significant, related to TC motion.

Line 254-255: “Relative magnitudes of the filamentation time are useful for providing a dynamic distinction between the inner and outer regions of TCs.” Has this been done in previous studies? If so, cite them.

Line 257: Please define MSL. Also, I believe that you really want “above mean sea level.”

Line 290: At this point, I was hoping for some commentary on the implications of your finding for the theory of initiation/propagation by gravity waves. It might be a good idea to make a note here in the text that you will explore the implications later in the section.

Line 299-302: This sentence seems to belong better with the previous paragraph. I suggest beginning the new paragraph with “Most of the observed outer TCRs ...”

Line 301: More information about whether principal rainbands are included in your dataset will help with interpretation here on the percentage of rainbands with “slow radial propagations.”

Line 355: Add citation to support use of $n=2$ for tangential wave number.

Line 393: Suggesting changing “the squall-line characteristics” to “the above outlined squall-line characteristics.”

Line 395: Please define “low-level,” “low-level band-normal vertical shear” and “band-relative band-normal wind component.” What levels and distances are used? How are the quantities calculated? Information can also be added to the Figure 7 caption.

Line 420-421: This is where I really started to get wary of your use of ERA-5 data for these mesoscale circulations associated with cold pool dynamics.

Line 428-429: This is a good way to account for some of the uncertainty. Unfortunately, due to data quality issues, I think the uncertainty is much larger.

Line 434-444: “that there is a much higher percentage of the squall line TCR group compared to that of the non-squall-line TCR group.” I don't follow, do you mean a higher percentage of overlap? or do you mean the density of the lines in the plot? like you've squashed the data more in Figure 7c so that more cases fit into the plot?

Line 448: characters → characteristics

Line 448: “fast movement” For clarity, are you referring to faster motion (blue dots higher magnitude) in Figure 7c?

Line 448-450: “One might see a more rapidly propagating nature for convective systems if they evolve with the operation of squall-line dynamics.” More rapid compared with what?

Line 490: When you refer to the rainbands being inside versus outside the “storm size,” I don't think “storm size” is the word you want since you can't really be inside the size. Rather, the rainbands are in/outside the storm radius, storm perimeter, or potentially the storm area.

Line 512: “must have existed” → must also exist

Line 540-546: Please be sure that you are providing information about all the datasets you used in this study, including the JTWC best track data.

Figure 1: I am wondering if there is a better way than arrows to demarcate the rainband. Could you reduce the transparency of the rest of the radar reflectivity regions? so that the rainband stands out more.

Figure 2: Could you add percentages or fraction on the right axis to assist with interpretation? This would be especially helpful with comparing fraction of inside vs. outside TC rainbands cited in the text.

Figure 5a,b,d: For interpretation, I manually added vertical lines along $x=0$ so that I could distinguish positive and negative values more easily. I think it would be great to add these to the plots.

NCOMMS-23-16082
Responses to Reviewer#1

Reviewer #1 (Remarks to the Author):

Summary:

The authors present a very nice radar climatology of outer-core tropical cyclone rainbands from 95 tropical cyclones near Taiwan from 2002-2019. Tropical cyclone rainbands are an important hazard and therefore this extensive climatology of their development and evolution is of value to the community. The authors use this radar climatology to document that outer-core tropical cyclone rainbands, defined as existing at least three times the radius of maximum winds from the storm center, predominantly develop within the outer-core region and do not commonly propagate outward from the inner-core region. The authors also use ERA-5 reanalysis data to suggest that the outer-core rainbands often propagate in a manner consistent with squall line dynamics and not consistent with vortex Rossby waves or gravity waves. Despite some potential limitations from using the relatively coarse ERA-5 reanalysis data I believe this analysis is appropriate.

My concerns are primarily that (1) I believe the authors misuse the word ‘origin’ by not providing a convincing description of what initiates the outer-core rainbands, (2) a lack of discussion on the diurnal cycle which is well established as influence tropical cyclone convection, and (3) a lack of discussion on vertical shear. Regarding (1), while I believe the authors adequately show that the convection associated with the outer-core rainbands develops within the outer-core region of the tropical cyclone, they do not provide a clear mechanism for their initiation. Instead, I believe the authors incorrectly use squall-line dynamics as an explanation, which I believe to instead be an explanation for their maintenance once convection is ongoing, but not their formation. With these concerns addressed, I believe the study can provide a useful climatology of tropical cyclone rainbands to the community.

Reply: We appreciate Reviewer#1's positive and constructive comments, which help us improve the manuscript. A set of responses to your detailed comments is provided below.

Major Comments:

1) I am concerned by the title of the paper and the use of the word 'origin' throughout the article. In my opinion the authors do not clearly show the origin of outer tropical cyclone rainbands. Rather I believe the authors demonstrate that the outer rainbands are often maintained through cold pool dynamics, not initiated by them. I fail to see the authors clearly demonstrate what initiates the outer rainbands. As a result, I believe the authors overstate their results such as in L44-46, L245-247, L503-504, L508-510. I also am not entirely convinced that the authors can rule out the role of gravity waves, vortex Rossby waves, or something else in triggering the initiation of the outer rainband convection before the cold pool dynamics help to support their maintenance thereafter (L497-502).

Reply: In this study, we investigate the statistical aspects of where the outer TCRs begin within the TC circulations (i.e., spatial origin) and of how the outer TCRs forms (i.e., mechanism origin) based on a large set of the outer TCRs observed during their formative stages. As for the spatial origin, we have shown the dominance of outer origin for the observed outer TCRs and this finding is well supported by our analyses (Figs. 3 and 4). As for the mechanism origin, although our analysis results have suggested the squall-line dynamics as an important mechanism for a large portion of the observed outer TCRs during their formative stage, we agree with the Reviewer's point that some more details about the mechanism origin need to be elaborated. To accommodate this issue, additional evidence is provided in the response here and in the revised manuscript to better illustrate how squall-line dynamics relate to the initiation of outer TCRs. As described in the Squall-line dynamics section of our original manuscript, during the preformation and formation stage of an outer TCR, its associated precipitation is inherently weak, with a very limited horizontal extent (Figs. 2a,b). The cooling processes accomplished by the nascent TCR itself should be generally ineffective, and there must be other preexisting precipitation separating from the considered outer TCR, which could act as an initial, critical provider of cold pools to initiate the operation of squall-line dynamics and contribute to the genesis of the rainband. In this context, the initiation of outer TCRs is expected to preferentially take place adjacent to some certain precipitation areas of TCs where evaporative cooling and/or convective transport is potentially active.

To validate the speculation above, an effort is made to construct the spatial composite of precipitation frequency (>15 dBZ) with respect to the location of each outer TCR at three particular time periods, 1 h prior to the preformation time (t_0-2), the preformation time (t_0-1) and the formation time (t_0). The preformation time is defined as the earliest, unambiguous detection of TCRs by radar observations

whereas the formation time refers to a time satisfying the formative criteria, as explained in the Data and rainband identification section. In this analysis, we choose the squall-line TCR cases with faster propagation velocities (i.e., greater than the mean cross-band propagation velocities of 6.1 m s^{-1} calculated from the squall-line TCR group) and corresponding to the theoretical velocity range of atmospheric cold pools to maximize composite signals for the squall-line outer TCRs. Similarly, those non-squall-line TCR cases that propagate relatively slower (i.e., less than the mean cross-band propagation velocities of 3.5 m s^{-1} calculated from the non-squall-line TCR group) and beyond the theoretical velocity range of atmospheric cold pools are chosen for the non-squall-line composite. As shown in the composite results, the position of the rainband is clearly marked by the highest precipitation frequency centered at $(x=0 \text{ km}, y=0 \text{ km})$ at t_0-1 and t_0 for both the squall-line TCRs and non-squall-line TCRs (Fig. R1). For the squall-line TCR composite, there is a concentrated area of preexisting precipitation on the rear flank of the rainband prior to its formation at t_0-2 (Fig. R1a). In particular, the rainband tends to be initiated immediately adjacent to the outer boundary of the preexisting precipitation in the outer regions of TCs (Figs. R1b,c), consistent with our aforementioned speculations. For the non-squall-line TCR composite, there are generally scattered distributions of lower precipitation frequencies (18-30%) over a broad region in both front and rear sides of the rainband (Figs. R1d,e,f). These features suggest that the rainband formation occurs in a relatively precipitation-free area and is not frequently associated with prominent preexisting precipitation. The contrasting statistical distribution of outer precipitation between the squall-line and non-squall-line TCR composites supports that the preexistence of suitable precipitation in the outer region of TCs would be critical to provide important sources of cold pools to initiate the operation of squall-line dynamics for the squall-line outer TCRs. A schematic diagram, as shown in Fig. R2, is also prepared to help illustrate the two distinct types of outer precipitation and their associated formative scenarios of squall-line and non-squall-line outer TCRs. In response to the Reviewer's comment, the above discussions along with Figs. R1 and R2 (i.e., Figs. 8 and 9) have been included in the Squall-line dynamics section of the revised manuscript.

On a different note, given that the lack of consistency in the propagation characteristics between the observed outer TCRs and VRWs/GWs, as elaborated in the Propagation and geometry characteristics section, it seems unlikely that the VRWs/GWs play a direct role in triggering the outer TCRs. Because the generation of some convection/precipitation within TCs would be probably related to these wave activities, we cannot entirely rule out the wave-induced precipitation as a potential

contributor that would favor the formation of cold pools and then initiate the operation of squall-line dynamics. To evaluate this likelihood and its underlying processes is beyond the scope of the study, but the potential, indirect roles of wave activities in facilitating the formation of outer TCRs deserve future investigation through detailed case studies using high-resolution observations and numerical simulations. In response to the Reviewer's comment, we have also added these discussions in the Squall-line dynamics section of the revised manuscript.

Fig. R1. The spatial composite of precipitation frequency (>15 dBZ, color shading) and normalized radial distance by RMW (contour) with respect to the location of each outer TCR ($x=0$ km, $y=0$ km) for 303 squall-line TCR cases at (a) 1 h prior to the preformation time (t_0-2), (b) the preformation time (t_0-1) and (c) the formation time (t_0). (d)-(f) as in (a)-(c), but showing the composite results for 140 non-squall-line TCR cases. For the composites at t_0-1 and t_0-2 in (a), (b), (d) and (e), the location of each TCR case is determined by the backward extrapolation from the TCR location at the formation time (i.e., t_0) based on the calculated propagation velocities of TCRs.

Fig. R2. Schematic diagram illustrating the formative scenario for the squall-line and non-squall-line outer TCRs, and their embedded precipitation characteristics are particularly highlighted. The solid circle denotes the inner-core boundary of TCs.

2) I understand that the authors used available radar data from Taiwan, but a natural question from readers will be how applicable are these results to tropical cyclones in other basins?

Reply: We thank the Reviewer for raising the valid question. Despite the different evolving stages of outer TCRs, the cold pool signatures associated with outer TCRs have been reported for TCs developing over different basins, such as the northwestern Pacific Ocean (e.g., Yu and Tsai 2013²⁸; Yu et al. 2018²⁹) and Atlantic Ocean (e.g., Powell 1990b⁵³; Eastin et al. 2012⁵⁵). In addition, outer TC circulations observed over different basins seem to have similar characteristics of low-level winds with the presence of some band-normal vertical shear (e.g., Powell 1990a⁵²; Robe and Emanuel 2001⁵⁴; Yu et al. 2018²⁹). It is thus reasonable to expect that what we learn from the study (i.e., the outer TCR origin and the importance of squall-line dynamics on the formation of outer TCRs) would be generally applicable to outer TCRs developing over other basins. However, a comprehensive dataset of outer TCRs collected from different geographical locations during their formative stage will be still required to clarify whether any possible basin-dependent characteristics of the TCR origin would exist. In response to the Reviewer's comment, we have added these points in the Summary section of the revised manuscript.

3) One aspect missing from the introduction and the analysis is the diurnal cycle, which is well known to influence tropical cyclone convection beyond the inner-core region (e.g., Dunion et al. 2014). The authors should consider if there is a diurnal cycle to outer-rainband formation and if the diurnal cycle may be an important mechanism to their formation.

Dunion, J. P., C. D. Thorncroft, and C. S. Velden, 2014: The Tropical Cyclone Diurnal Cycle of Mature Hurricanes. *Mon. Wea. Rev.*, 142, 3900 – 3919, <https://doi.org/10.1175/MWR-D-13-00191.1>.

Reply: Thanks to the Reviewer for bringing this issue to our attention. In this revision, we have included the aspect of diurnal cycles in the Introduction section, and relevant articles have also been cited. In addition, the formation times of all outer TCRs observed in this study are also analyzed to understand if there are any diurnal signals. The statistical distribution, as shown in Fig. R3, indicates that the number of outer TCRs formed at different times in a day is similar and approximately equal to 80 cases for each time interval, although the peak and lowest numbers of TCR formation are found afternoon between 12 and 14 pm and between 14 and 16 pm, respectively. This result generally reflects the lack of obvious diurnal variations for the formation of the studied outer TCRs. In this revision, we have included these descriptions in the Formative location and dynamic regime section of the revised manuscript for clarity.

Fig. R3. Formative number of the outer TCR cases observed in this study at different time intervals during a day.

4) A second aspect that appears to be missing from the analysis is a discussion on

vertical wind shear, which is well known to influence tropical cyclone convection (e.g., Corbosiero and Molinari, 2002). I recommend adding an additional panel (Figure 3) that shows the formation locations relative to the deep-layer shear in an RMW relative coordinate. The criteria mentioned for the classification of an outer tropical cyclone rainband required the rainband forming or propagating into the region beyond 3 times the RMW. Showing the formation locations relative to the RMW and shear may yield additional insights.

Corbosiero, K. L., and J. Molinari, 2002: The Effects of Vertical Wind Shear on the Distribution of Convection in Tropical Cyclones. *Mon. Wea. Rev.*, 130, 2110 – 2123, [https://doi.org/10.1175/1520-0493\(2002\)130<2110:TEOVWS>2.0.CO;2](https://doi.org/10.1175/1520-0493(2002)130<2110:TEOVWS>2.0.CO;2).

Reply: Thanks to the Reviewer for bringing this issue to our attention. In this revision, we have included the aspect of large-scale vertical wind shear in the Introduction section, and relevant articles have also been cited. We also agree with the Reviewer that the modulation of the outer TCR formation by deep-layer vertical wind shear is likely to occur. However, as described in the Formative location and dynamic regime section, the azimuthal variation of the outer TCR formation observed in this study is contaminated by the inherent limitation of radar observations. Specifically, the precipitation information of the front quadrants for a typical westward-moving TCs over the northwestern Pacific Ocean could be usually monitored by coastal radars in Taiwan for a longer period of time. In contrast, when the rear quadrants of TCs reach the observational coverage of coastal radars, the inner-core circulations of TCs have been closer to the landmass of Taiwan so both TC circulations and rainbands have greater chances to experience orographic modifications and/or make landfall. This sampling preference of coastal radars is expected to capture more outer TCR cases in the front quadrants than in the rear quadrants, consistent with the asymmetric characteristics of formative locations of the observed outer TCRs, as shown in Fig. R4a (i.e., Fig. 3 of the original manuscript).

In addition, a plan view of formative locations for all analyzed outer TCRs with respect to the large-scale deep-layer vertical shear is shown in Fig. R4b. The analysis seems to indicate that outer TCR formation would occur more frequently in the upshear quadrants than in the downshear quadrants. However, this asymmetric distribution is also most likely contributed by the sampling preference of coastal radars as discussed above, in which the directions of large-scale deep-layer vertical shear are typically opposite to the motion vectors of TCs. As shown in Fig. R5, most of the studied TCR cases (~70%) have large differential angles between the TC motion vectors and shear vectors greater than 90° (i.e., obtuse angles). The mean directions

for the TC motion vector and the shear vector are calculated to be 317° and 195° , respectively. These statistical characteristics support the substantial contribution of the radar sampling to the asymmetric distribution shown in Fig. R4b. This is the reason why we do not attempt to address the outer TCR formation in a spatial coordinate relative to the TC motion and/or the deep-layer vertical shear. In the future, effective mitigation in the bias of radar sampling, such as by collecting long-term observations from isolated island radars located over the open ocean, will be critically required to document the natural variations of outer TCR formation over different quadrants of TCs. For clarity, the inherent limitation of radar sampling and its potential impact on the observed asymmetric distribution of outer TCR formation have been explained in more detail in the Formative location and dynamic regime section of the revised manuscript.

Fig. R4. (a) Plan view of formative locations for all identified outer TCRs (1029 cases) relative to the TC center and motion. The TC motion is toward the top of the page. The range rings are indicated every 100 km. (b) Same as (a), but showing the formative locations with respect to the large-scale deep-layer vertical shear that is calculated between 700 and 200 hPa within a radial distance of 150-500 km from the ERA5 reanalysis data.

Fig. R5. Number of outer TCR cases at different intervals of differential angles. The differential angle is defined as the angle between the TC motion vector and deep-layer shear vector.

5) I think the use of ERA5 for the TC presents potential for criticism. This may be unavoidable, but the authors should add some discussion from using $.25^\circ$ data to represent the TC conditions and the associated gravity wave and vortex Rossby wave calculations within their analysis (L370-373, L432-433). As an example, can we trust filamentation time estimated from relatively coarse ERA5 data?

Reply: We thank the Reviewer for raising the concern with the ERA5 reanalysis data. As described in the Data and rainband identification section, because ERA5 data have higher spatial and temporal resolutions compared to other reanalysis data, it has been widely used to study relevant scientific issues for TCs and their environment. However, we agree with the Reviewer that whether ERA5 data can provide adequate representation for the low-level winds nearby the studied outer TCRs is somewhat uncertain. To fully address this issue is beyond the scope of this study, but an effort is made to evaluate this uncertainty in this study by comparing the ERA5 winds with other independent wind measurements nearby the rainband's location. In making this comparison, observations from eight offshore island stations [Pengjiayu (PJY), Yonaguni (YO), Hateruma (HTM), Iriomote (IM), Ishigaki (IG), Green island (GI), Penghu (PH) and Dongji (DJ)] located near Taiwan (Fig. R6) within a distance of 35 km from the formative location of outer TCRs are first collected. The reason why the threshold distance of 35 km is used herein is because it is approximately comparable to the grid spacing of ERA5. The island observations represent the "ground truth" measurements and are available from a total of 70 outer TCR cases. Compared to the

large number of outer TCRs analyzed in the study, this is a relatively small set of TCR cases, but they are valuable to provide a critical assessment of the representation of the ERA5 data in the rainband environment. The 10-m (MSL) winds from the ERA5 gridded data closest to these island stations for these TCR cases are then chosen to compare with station-measured winds. Here, we focus on the comparison of the tangential winds [i.e., V in Eqs. (5), (6) and (7)] and band-normal winds [i.e., V_c in Eq. (8)] because they are the ERA5-provided kinematic quantities involved with the calculation of VRWs and the evaluation of squall-line dynamics.

Figure R7a shows the scatterplot of tangential wind components calculated from the ERA5 data and offshore island observations. The result indicates a good correlation between the two datasets, and the correlation coefficient, average error and root mean square error (RMSE) are calculated to be 0.74, 2.9 m s^{-1} and 4.5 m s^{-1} , respectively. If we assume the calculated RMSE as representative errors for the tangential winds from ERA5, potential velocity errors in radial and tangential directions can be estimated to be 0.2 and 4.4 m s^{-1} , respectively, based on Eqs. (2)-(7). Figure R7b shows the scatterplot of band-normal wind components calculated from the ERA5 data and offshore island observations. Similar to the tangential winds, a good correlation is also evident for the band-normal winds. The correlation coefficient, average error and root mean square error (RMSE) are calculated to be 0.77, -0.1 m s^{-1} and 2.9 m s^{-1} , respectively. The value of the calculated RMSE yields an error of 1.7 m s^{-1} for the estimate of theoretical cold-pool velocity based on Eq. (8). The potential errors in the calculated propagation velocity for VRWs and cold pools, as estimated above, may be considered relatively minor because they are unlikely to alter the statistical distributions of the scatterplots shown in Figs. 6b,c and Fig. 7c,d significantly.

The error analyses above generally support the reasonable representation of the ERA5 reanalysis data for the rainband environment, although the accuracy of the ERA data to represent TC circulations and their ambient wind conditions appears to vary case by case (Fig. R7). This uncertainty may have greater chances to become a potentially serious problem if the scientific approach is placed on the understanding of individual TC/TCR events. However, because the present study focuses on the statistical aspects of outer TCR formation using long-term reanalysis data for a very large set of TCR cases, the potential impact of this data uncertainty may be appropriately mitigated. In response to the Reviewer's comment, we have included discussions given above in the Propagation and geometry characteristics section and Squall-line dynamics section in this revision.

Fig. R6. Radar dataset used in this study. The locations of the coastal Doppler radars at Wu-Fen-San (WFS), Hua-Lien (HL), Chi-Gu (CG) and Ken-Ting (KT) are denoted by triangles. The locations of offshore island observations near Taiwan at Pengjiayu (PjY), Yonaguni (YO), Hateruma (HTM), Iriomote (IM), Ishigaki (IG), Green island (GI), Penghu (PH) and Dongji (DJ) are marked by solid circles. The outer curve denotes the approximate data coverage for these radars, extending from the island of Taiwan to 300–400 km offshore.

Fig. R7. (a) Scatterplot of the tangential wind components (V_t) calculated from the ERA5 data and offshore island observations. (b) Same as (a), but showing the statistical results of the band-normal wind components (V_c).

Minor Comments:

L41: I recommend adding “radar” before “observations” .

Reply: Revised as suggested.

L53: “storm” should be changed to “storms” .

Reply: Corrected as suggested.

L81: I think “inner-core” should be “outer-core” .

Reply: The reason why “inner-core” is stated here is because we would like to emphasize a possible formative scenario, in which outer TCRs would be initially triggered by gravity waves within the inner-core region and then propagate outward to the outer regions of TCs. This formative scenario of outer TCRs is so-called “inner origin” as described in the Introduction section. In this revision, this sentence has been reworded for clarity.

L94: “would probably” can be changed to “could” and “but” can be removed.

Reply: Reworded as suggested.

L100: “actually” can be removed.

Reply: Removed as suggested.

L102: “more” can be removed.

Reply: Removed as suggested.

L127: “evidence” can be removed.

Reply: Removed as suggested.

L184-186: It should be mentioned how the orographically generated rainbands were removed from the analysis.

Reply: Thanks for raising this valid point. One of the most common features of the topographically generated rainbands is that they are oriented approximately parallel

to the eastern coast of Taiwan without obvious curvature. In addition, these rainbands tend to be locked with coastal mountains and do not propagate rapidly with cyclonic circulations of TCs. These characteristics contrast sharply with evolving and propagation aspects of TCRs. Hence, if rainbands are observed to form locally within the coastal zone (from coastline to ~25-100 km offshore) and meet the above signatures, they are assumed to be the topographically generated rainbands and are thus removed during TCR identification. For clarity, these explanations have been added in this revision.

L233: Do the authors mean this is a result of the Taiwan radars more frequently sampling the front quadrants as tropical cyclones approach Taiwan?

Reply: Sorry for the confusing description. This part of the text has been significantly rewritten for clarity. Please also see our responses to Major comment#4.

L264-268: The size of a tropical cyclone is often considered to be larger than the radius of tropical storm force winds. As mentioned by the authors other metrics are also commonly used including the outermost closed isobar and the radial extent of weaker wind speed thresholds, such as 10 or 12 m/s. The authors should be careful in defining tropical cyclone size as the radial extent of tropical storm force winds (34 kts). See also L490.

Reply: Thank the Reviewer for raising this good point. In this revision, we explicitly define R34 as "storm radius", instead of simply using the term "TC size", to avoid any possible ambiguity. For clarity, this part of the text has been reworded as below:

"Since outer TCR formation is observed to be active over extensive outer regions of TCs, as shown in Figs. 3 and 4a–c, one intriguing question emerges, what is the degree of prevalence for outer TCRs to form at radii larger than the storm radius of TCs? The size of a TC has been practically defined as the radius of a certain threshold of near-surface wind speed or by the radial extent of the outermost closed isobar⁷³⁻⁷⁵. Because the TC size may vary considerably with different thresholds chosen, we use the average radius of 34 kt winds (R34) as a storm radius for TCs in this study. R34 is commonly considered as the area of the TC warnings issued by operational forecasting centers³¹. The statistical distribution of the TCR formative number, as a function of normalized radial distance by JTWC-recorded R34, is shown in Fig. 4d. The peak outer TCR formation (139 cases) is observed within the interval of normalized radial distances between 0.75 and 1, which is close to the outer boundary of the

storm radius of R34. Although the formative numbers tend to decrease generally with increasing radial distance in regions beyond the storm radius, the formative fractions of outer TCRs inside and outside the storm radius are dramatically different, which are calculated to be 29% and 71%, respectively. These statistics further indicate that the majority of outer TCR formation actually occurs in far outer regions beyond the operational TC alert area.”

L283: I recommend adding “near Taiwan” following “outer TCRs” .

Reply: Revised as suggested.

L284: “crossing angles” needs to be defined somewhere.

Reply: The definition of crossing angles has been given in the Introduction section as below:

“The crossing angle is defined as the angle between the rainband’s orientation and the tangent to a circle with radius from the band to the TC center.”

L357: “actually” can be removed.

Reply: Removed as suggested.

L386: “circumstances to activate the operation” can be removed.

Reply: Thanks for the suggestion. For clarity, this part of the sentence has been reworded as “present at low levels are two of the most fundamental characteristics of squall-line dynamics.”

L462,510: I think the authors mean for “predominantly” to be “exclusively” .

Reply: Revised as suggested.

L465,504,519: The authors show that squall-line dynamics are often important for the maintenance of the rainbands, not the “formation” .

Reply: Thanks for raising the valid point. Please see our responses to Major comment#1 for details.

NCOMMS-23-16082
Responses to Reviewer#2

Reviewer #2 (Remarks to the Author):

Review of “Origin of outer tropical cyclone rainbands”

Synopsis:

This is an important study that utilizes a radar dataset to investigate the formation location and movement of outer tropical cyclone (TC) rainbands, as well as potential formation mechanisms. As discussed in the introduction, there are competing theories on the formation and origin of outer TC rainbands, with numerous (mostly modeling) studies indicating that these rainbands are generated as a result of TC inner core dynamic. In the current study, the authors provide convincing evidence that most outer TC rainbands form outside the inner core, and in some cases outside the radius of gale force winds. Additionally, the rainbands propagate at radial and tangential velocities that are inconsistent with inner origin theories. Instead, the authors show that a majority our outer TC rainbands propagate in a manner consistent with squall line (and cold pool) dynamics. Therefore, I believe that this study is an important contribution to the research literature. However, I have major concerns associated with data and methods, which are outlined below. Specifically, use of ERA5 data in certain parts of the study is not justified and not likely to be sound. As a result, I recommend major revisions for this manuscript.

Reply: We appreciate Reviewer#2’s positive and detailed comments, which help us improve the manuscript. A set of responses to your detailed comments is provided below.

Major comments:

1. Use of ERA-5 data: On lines 158-163, the authors provide some basic information about the ERA-5 dataset and justify its use for it “more realistic representation of TCs and their environment.” It is generally true that the ERA-5 dataset provides more realistic data about the TC environment as well as some other TC-related metrics such as TC size. However, in this study, the authors have used ERA-5 data to capture information about lower-level (boundary layer) winds that are unlikely to be accurate due to the low resolution of the associated ERA5 model and the need for these winds to be calculated through parameterization. Could the authors find another dataset that would provide more accurate low-level wind info? Later in the

results, the authors use ERA5 data to assess whether cold pool dynamical (and thermodynamical) processes are present, and these mesoscale processes are very unlikely to be captured by the low-resolution ($0.25^\circ \times 0.25^\circ$) ERA5. I have major concerns about using ERA5 to assess mesoscale squall-line processes.

Reply: We thank the Reviewer for raising the concern with the ERA5 reanalysis data. As described in the Data and rainband identification section, because ERA5 data have higher spatial and temporal resolutions compared to other reanalysis data, it has been widely used to study relevant scientific issues for TCs and their environment. However, we agree with the Reviewer that whether ERA5 data can provide adequate representation for the low-level winds nearby the studied outer TCRs is somewhat uncertain. To fully address this issue is beyond the scope of this study, but an effort is made to evaluate this uncertainty in this study by comparing the ERA5 winds with other independent wind measurements nearby the rainband's location. In making this comparison, observations from eight offshore island stations [Pengjiayu (PJY), Yonaguni (YO), Hateruma (HTM), Iriomote (IM), Ishigaki (IG), Green island (GI), Penghu (PH) and Dongji (DJ)] located near Taiwan (Fig. R8) within a distance of 35 km from the formative location of outer TCRs are first collected. The reason why the threshold distance of 35 km is used herein is because it is approximately comparable to the grid spacing of ERA5. The island observations represent the "ground truth" measurements and are available from a total of 70 outer TCR cases. Compared to the large number of outer TCRs analyzed in the study, this is a relatively small set of TCR cases, but they are valuable to provide a critical assessment of the representation of the ERA5 data in the rainband environment. The 10-m (MSL) winds from the ERA5 gridded data closest to these island stations for these TCR cases are then chosen to compare with station-measured winds. Here, we focus on the comparison of the band-normal wind [i.e., V_c in Eq. (8)] because it is the only ERA5-provided kinematic quantity involved with the evaluation of squall-line dynamics.

Figure R9 shows the scatterplot of band-normal wind components calculated from the ERA5 data and offshore island observations. The result indicates a good correlation between the two datasets, and the correlation coefficient, average error and root mean square error (RMSE) are calculated to be 0.77, -0.1 m s^{-1} and 2.9 m s^{-1} , respectively. If we assume the calculated RMSE as a representative error for the band-normal winds from ERA5, the potential velocity error in the calculation of theoretical cold-pool velocity based on Eq. (8) can be estimated to be 1.7 m s^{-1} . This potential error may be considered relatively minor and is unlikely to alter the statistical distributions of the scatterplots shown in Figs. 7c,d significantly.

The error analyses above support the reasonable representation of the ERA5 reanalysis data for the rainband environment, although the accuracy of the ERA data to represent TC circulations and their ambient wind conditions appears to vary case by case (Fig. R9). This uncertainty may have greater chances to become a potentially serious problem if the scientific approach is placed on the understanding of individual TC/TCR events. However, because the present study focuses on the statistical aspects of outer TCR formation using long-term reanalysis data for a very large set of TCR cases, the potential impact of this data uncertainty may be appropriately mitigated. In response to the Reviewer's comment, we have included discussions given above in the Propagation and geometry characteristics section and Squall-line dynamics section in this revision.

Fig. R8. Radar dataset used in this study. The locations of the coastal Doppler radars at Wu-Fen-San (WFS), Hua-Lien (HL), Chi-Gu (CG) and Ken-Ting (KT) are denoted by triangles. The locations of offshore island observations near Taiwan at Pengjiayu (PJY), Yonaguni (YO), Hateruma (HTM), Iriomote (IM), Ishigaki (IG), Green island (GI), Penghu (PH) and Dongji (DJ) are marked by solid circles. The outer curve denotes the approximate data coverage for these radars, extending from the island of Taiwan to 300–400 km offshore.

Fig. R9. Scatterplot of the band-normal wind components (V_c) calculated from the ERA5 data and offshore island observations.

2. Please provide more information about your “subjective screening” of the radar data to locate outer TC rainbands.

Reply: In this revision, we have included more explanations on the method to identify an outer TCR. Please see responses to Major comment#2a below for details.

a. As described in the text, the first set of criteria include measurements of the rainband and other quantitative criteria, which suggest that it could have been applied algorithmically. Or was this part applied manually too? If manual, please provide justification for doing so. Manual/subjective application of the length-to-wind-ratio and minimum length seems less than ideal.

Reply: Sorry for the confusion. For clarity, this part of the text has been rewritten to include more detailed information for the identification of outer TCRs, as described below:

“To seek TCRs, a very large set of radar reflectivity images and data from the lowest plan position indicator (PPI) scans (elevation $\sim 0.4^\circ$) for TCs approaching Taiwan is initially prepared. The dataset comes from the four radars and contains observations of 95 TCs during 2002–2019, which the CWB issued typhoon warnings. All of the reflectivity images produced are first screened visually to determine potential candidates of TCRs whose radar echoes can be tracked backward from their mature to formative stages. The primary purpose of this subjective screening is just to retain

TCR cases that could be captured successfully by radars during their initiation time. Reflectivity data from these tentatively chosen TCR cases are then calculated to obtain basic rainband information, such as location, length and width, for each of the TCRs. The specific location of rainbands at a given time is determined by averaging the azimuthal angles and radial distances with respect to the TC center over the rainband entity with reflectivities ≥ 30 dBZ. The length of rainbands is determined by the distance between two ends of the long axis of the rainband entity with reflectivities ≥ 20 dBZ. Similarly, the width of rainbands is approximated by a mean distance in the band-normal direction along the rainband entity. With these calculated rainband characteristics above, it is possible to identify an outer TCR based on objective, quantitative criteria. The criteria used to identify a TCR are similar to those used by previous TCR studies^{9,29} but more specific criteria are used herein to define an outer TCR and determine its initiation. First, a precipitation feature with a maximum reflectivity of at least 30 dBZ, a minimum length of 100 km and a large length-to-width ratio (>5) must be satisfied. Second, the precipitation signature described above must persist for 1 h or longer. These two criteria can assure that a prominent and persistent rainband has been identified, and that a relatively transient, non-banded or loosely organized precipitation feature could be effectively avoided. Third, a rainband is classified as an outer TCR as long as its precipitation occurs in or propagates into regions beyond a radial distance of 3 times the RMW for a certain time period during rainband formation and development. This criterion allows us to include both possible formative scenarios: one is that the rainband begins development within the inner TC core and then propagates outward over time into the outer region, and the other is that the rainband is initiated locally in the outer region. Fourth, the formation time for an outer TCR is defined as when the rainband initially satisfies the first criterion above.”

b. Do your data include the principal rainband? or have you excluded it using your subjective analysis?

Reply: In this study, we do not attempt to exclude principal bands using subjective analysis. Identification of all outer TCRs analyzed in this study are based on the objective, quantitative criteria, as described in our responses to Major comment#2a above. An observed rainband is considered as an outer TCR as long as it satisfies these criteria. So, we cannot rule out the possibility that the outer TCR dataset may contain principal rainbands.

c. Lines 184-186: Please elaborate on how you identified orographically generated

rainbands.

Reply: Thanks for raising this valid point. For clarity, we have added explanations for the identification of orographically generated rainbands in this part of the text, as below.

“One of the most common features of the topographically generated rainbands is that they are oriented approximately parallel to the eastern coast of Taiwan without obvious curvature. In addition, these rainbands tend to be locked with coastal mountains and do not propagate rapidly with cyclonic circulations of TCs. These characteristics contrast sharply with evolving and propagation aspects of TCRs. Hence, if rainbands are observed to form locally within the coastal zone (from coastline to ~25-100 km offshore) and meet the above signatures, they are assumed to be the topographically generated rainbands and are thus removed during TCR identification.”

d. Lines 236-237: “Nevertheless, considerable TCRs are observed to form in far outer TC regions beyond the radius of 350 km.” How do you identify whether these rainbands are associated with the TC rather than some other forcing? Please provide more details in the methods to describe how you decipher whether the rainband is associated with the TC. Rainbands at distance ~1000 km from center seem like they could be forming due to some other mechanism.

Reply: Thanks for raising the question and good point. As described in the “Data and rainband identification” section, the identification task for outer TCRs is focused only on the time period with the official typhoon warnings issued by Central Weather Bureau (CWB) of Taiwan. This ensures that TCs would be a primary circulation system influencing the Taiwan area and its surrounding ocean area. Moreover, all of the outer TCRs identified by radar observations in this study have also been checked if they are mainly influenced by TC circulations and are not associated with any other synoptic weather systems (e.g., fronts or cyclones). This checking is practically done by looking at both weather analysis maps provided from CWB of Taiwan and atmospheric circulations nearby TCs seen from the ERA5 reanalysis data.

In addition, as the Reviewer kindly mentioned, rainbands at far distances from the TC center are also evident. Indeed, these exceptionally distant rainbands are quite rare; only a single rainband located beyond 1000 km and 11 rainbands (~1 %) located between 800-1000 km (Fig. 3 and Fig. 4a). With the examination of CWB weather

analysis maps (not shown), we can initially preclude the possibility for the influences of these outer TCRs by other synoptic weather systems. Also, as seen from the ERA5 data, these TCR cases are primarily associated with cyclonically rotational circulations in the outer region of TCs, as shown in Fig. R10. Based on these observational facts, it is reasonable to conclude that these distant TCR cases belong to precipitation features exclusively induced by TC forcings.

In response to the Reviewer's comment, we have included the explanations above in the Data and rainband identification section of the revised manuscript for clarity.

Fig. R10. The spatial composite of horizontal winds at 1.5 km MSL (streamline) with respect to the TC center from the ERA5 data for 12 outer TCR cases with radial distances beyond 800 km. Formative locations of these TCR cases are marked by red stars for reference.

Minor comments:

Line 37: I don't agree on the outer rainbands being the "most persistent." The principal rainband is more persistent. It might be better to focus on the hazardous weather part of this statement (or the extreme rain rates observed in these rainbands). For example, you mention in the article that these rainbands can exist outside the TC gale force wind radius.

Reply: Thanks for the valid point. Currently, there is no relevant research to document the statistical differences in the duration between individual principal bands and outer TCRs observed in the real atmosphere. Although our experiences learned from looking at numerous TCs approaching Taiwan indicate that the

appearance of principal bands within TCs seems less frequent than outer TCRs, we agree with the Reviewer that saying “most persistent” for outer TCRs may be overstated. In this revision, we have removed the description of “most persistent” and this part of the text has also been reworded to emphasize the aspects of outer TCRs that are a concentrated region of heavy precipitation and hazardous weather within tropical cyclones (TCs).

Line 45: squall line dynamics “are” a common mechanism.

Reply: Revised as suggested.

Line 46: I don’t think the word “road map” is the right choice because you don’t specifically outline research directions and how those should be navigated.

Reply: For clarity, the “road map” and its associated sentences in the Abstract and Summary sections have been removed in this revision.

Line 54: There is not a clear connection between sea level rise and loss of life (it’s leading to enhanced storm surges that lead to loss of life). Suggest rewording.

Reply: In response to the Reviewer’s comment, this sentence has been reworded for clarity.

Line 64-68: I disagree with characterization of inner vs. outer rainbands as transient (inner) and persistent (outer). Inner rainbands are usually dynamically driven and therefore persist (e.g., eyewall, principal rainband) whereas outer, convective rainbands are convective (thermo-driven) and less persistent in terms of their location and duration. Of course, some of the inner rainbands within the stationary band complex are more transient (such as secondary rainbands), but the eyewall and principal rainband in a major hurricane may be the most persistent and defining features of these storms. Some of the articles you cite (e.g., Willoughby et al. 1984) point out the persistent, stationary nature of the principal rainband. Convective storms can persist for hours but not over multiple days as the eyewall and principal rainband do.

Reply: We agree with the Reviewer that the precipitation feature of eyewalls could be more persistent compared to outer TCRs. However, tropical cyclone rainbands (TCRs) defined in the literature usually refer to spirally banded features of

precipitation (i.e., spiral rainbands) outside the eyewall. TCRs are conventionally classified into inner and outer TCRs based on their location relative to the TC center. Therefore, the terminology of inner TCRs usually excludes the rainband type of eyewalls, although the eyewall is actually located within the inner-core region of TCs. In addition, regarding the principal bands, we choose to follow our responses to Minor comment # Line 37 above, and the “persistent” and “transient” words have been both removed from this part of the text for clarity.

Lines 68-70: This sentence is about the impacts on human and natural communities, but the references (28,29,31,32) are mostly about the structure of outer TC rainbands. The only cited article that seems relevant here is the one about meteotsunamis. I suggest finding studies that focus on rainband/flooding impacts instead.

Reply: Thanks for raising this concern. The references 28 and 29 cited provided strong, comprehensive observational evidence of squall-line-like precipitation and airflow signatures (i.e., severe weather signals) associated with outer TCRs. The reference 31 described analysis results about how outer TCRs contributed to the occurrence of hazardous weather and how they impacted a fatal airline crash. The reference 32 provided evidence of how TCRs trigger meteotsunamis. Therefore, the four references appear relevant to the statement herein. In response to the Reviewer’s comment, one more reference related to severe flooding due to the interactions of landfalling TCRs with topography is also cited herein.

Line 71: I think you might want to cite more or different studies here, at the very least Part II by Moon and Nolan (2015), which I believe is the more relevant study about the formation and propagation of inner rainbands.

Moon, Y., and D. S. Nolan, 2014: Spiral Rainbands in a Numerical Simulation of Hurricane Bill (2009). Part II: Propagation of Inner Rainbands. *J. Atmos. Sci.*, 72, 191 – 215, <https://doi.org/10.1175/JAS-D-14-0056.1>.

Reply: Thanks for pointing this out. Yes, Moon and Nolan (2015) (Part II) above is a more relevant study and it has been cited in this revision.

Line 73: There are a few more recent studies about observations of inner rainbands and the VRW theory. Suggest citing some here, primarily:

Guimond, S. R., P. D. Reasor, G. M. Heymsfield, and M. M. McLinden, 2020: The Dynamics of Vortex Rossby Waves and Secondary Eyewall Development in Hurricane

Matthew (2016): New Insights from Radar Measurements. Journal of the Atmospheric Sciences, 77, 2349 – 2374, <https://doi.org/10.1175/JAS-D-19-0284.1>.

Reply: Thanks for recommending the nice article related to VRW/inner TCRs. This article has been cited herein in this revision.

Line 75: delete extra word “be”

Reply: Deleted as suggested.

Line 78: “Despite” doesn’t seem to be the right word choice here. Maybe “Due to” ? Think about rewording this sentence on lines 78-82 to better reflect the meaning you’re conveying.

Reply: For clarity, this sentence has been reworded for clarity.

Lines 93-95: I don’t follow how the principal rainband is related to outer or distant rainband formation. You say in the text “outer-origin scenarios would probably be related” to this rainband, but you don’t explain how. Could you elaborate? Or maybe there is no underlying theory and this sentence should be reworded or removed.

Reply: Given that there is no underlying theory, this sentence has been reworded and other relevant descriptions are also revised for clarity.

Lines 97-99: “Previous PB studies rely dominantly on the analyses of airborne flight-level and radar data collected over a relatively short period of time during the mature or late stage of oceanic TCRs.” There are multiple modeling and idealized studies of the principal rainband by Michael Riemer and colleagues. Please revise. Riemer, M., 2016: Meso- β -scale environment for the stationary band complex of vertically sheared tropical cyclones. Q.J.R. Meteorol. Soc., 142, 2442 – 2451, <https://doi.org/10.1002/qj.2837>.

Reply: Thank the Reviewer for bringing this article to our attention. We have cited this article in this revision and this part of the text is also reworded for clarity.

Lines 101-102: “To the best of our knowledge, Yu et al. (2019) is the only observational study of TCRs to provide a more complete picture of the outer-origin

processes.” This may be true. However, there are multiple studies that observe the cold pools associated with these rainbands and their squall-line like structure. I think you should also attribute the idea to these other studies, rather than citing them below as an afterthought (since the Yu et al. 2019 study came later and built on these former studies).

Eastin, M. D., T. L. Gardner, M. C. Link, and K. C. Smith, 2012: Surface cold pools in the outer rainbands of Tropical Storm Hanna (2008) near landfall. *Mon. Wea. Rev.*, 140, 471 – 491, doi:10.1175/MWR-D-11-00099.1.

Moon, Y., and D. S. Nolan, 2014: Spiral Rainbands in a Numerical Simulation of Hurricane Bill (2009). Part I: Structures and Comparisons to Observations. *J. Atmos. Sci.*, 72, 164 – 190, <https://doi.org/10.1175/JAS-D-14-0058.1>

Yu, C.-K. & Tsai, C.-L. Structural and surface features of arc-shaped radar echoes along an outer tropical cyclone rainband. *J. Atmos. Sci.* 70, 56 – 72 (2013).

Reply: This is a valid point. In this revision, we have re-organized this part of the text by moving the paragraph starting with “With advances....” to the position before the paragraph starting with “To the best of” to better reflect the subsequence of scientific thought.

Line 139: Can you provide a citation or website for these statistics about the number of landfalling TCs around Taiwan from the Taiwan CWB?

Reply: We have provided the website information (https://rdc28.cwb.gov.tw/TDB/public/warning_typhoon_list/) along with the description of the statistics.

Line 143-145: The observations cannot “aim.” Reword this sentence.

Reply: Reworded as suggested.

Line 165: Please provide a website or reference for the JTWC best track data. It can be provided here in the text, but you may want to actually provide this information in the data availability statement.

Reply: The website information for JTWC best track data has been provided in the Data availability statement, as the Reviewer suggested.

Line 167-169: “The dataset comes from the four radars and contains observations

of 95 TCs during 2002 – 2019, which the CWB issued typhoon warnings.” Here, you’ re using “which” in a relative clause, but I think it needs to be used in a preposition: “The dataset comes from the four radars and contains observations of 95 TCs for which the CWB issued typhoon warnings (during 2002 – 2019).”

Reply: Revised as suggested.

Lines 213-222: It's good to include these limitations. Could you provide even more information, such as the distribution of errors, not just the average value. It would be nice to see a table to histogram that summarizes this information more fully, since the average might obscure other important information about the full distribution, including very large error values due to a non-normal distribution.

Reply: We have prepared the statistical histogram for the differences between the JTWC-calculated and radar-estimated propagation velocities, as shown in Fig. R11. The differences are characterized by a normal distribution centered at nearly zero values ($\pm 1 \text{ m s}^{-1}$) for both radial and tangential propagation velocities. Although large differences (i.e., beyond $\pm 6 \text{ m s}^{-1}$) are also evident, their occurrence is rare, with much lower percentages. The statistics indicate a general consistency between the two independently estimated propagations. Given no perfect observation is available to estimate the TC motions, the differences discussed herein do not actually represent errors, and they just imply a possible uncertainty for the estimation of TC motions based on independent datasets and methods. In response to the Reviewer’s comment, more statistical information about the differences has been included in this revision.

Fig. R11. (a) The statistical histogram for the differences between the JTWC-

calculated radial propagation velocities and radar-estimated radial propagation velocities from the 174 outer TCR cases. (b) As in (a), but showing the results of the differences for the tangential propagation velocities.

Lines 232-233: “This weakly asymmetric characteristic may not be statistically robust and would be due in part to the relative configurations between the generally westward-moving TCs in the northwestern Pacific Ocean and the Taiwan area.” Elaborate and support this statement better. I don't understand why these would not be statistically significant, related to TC motion.

Reply: In response to the Reviewer's comment, more explanations have been added in this revision and relevant descriptions are also revised for clarity, as below:

“This azimuthal variation of the outer TCR formation, however, would be due in part to the contamination by the inherent limitation of radar observations^{9,29}. Specifically, the precipitation information of the front quadrants for a typical westward-moving TCs over the northwestern Pacific Ocean could be usually monitored by coastal radars in Taiwan for a longer period of time. In contrast, when the rear quadrants of TCs reach the observational coverage of coastal radars, the inner-core circulations of TCs have been closer to the landmass of Taiwan so both TC circulations and rainbands have greater chances to experience orographic modifications and/or make landfall. This sampling preference of coastal radars is expected to capture more outer TCR cases in the front quadrants than in the rear quadrants, consistent with the asymmetric characteristic of formative locations of the observed outer TCRs, as shown in Fig. 3.”

Line 254-255: “Relative magnitudes of the filamentation time are useful for providing a dynamic distinction between the inner and outer regions of TCs.” Has this been done in previous studies? If so, cite them.

Reply: Additional references have been cited here for clarity.

Line 257: Please define MSL. Also, I believe that you really want “above mean sea level.”

Reply: In this revision, “MSL” has been defined in this sentence for clarity.

Line 290: At this point, I was hoping for some commentary on the implications of

your finding for the theory of initiation/propagation by gravity waves. It might be a good idea to make a note here in the text that you will explore the implications later in the section.

Reply: A note is added right after the sentence, as the Reviewer suggested.

Line 299-302: This sentence seems to belong better with the previous paragraph. I suggest beginning the new paragraph with “Most of the observed outer TCRs ...”

Reply: Revised as suggested.

Line 301: More information about whether principal rainbands are included in your dataset will help with interpretation here on the percentage of rainbands with “slow radial propagations.”

Reply: As described in our responses to Major comment#2b, we cannot rule out the possibility that our outer TCR dataset may also contain principal rainbands. The observed outer TCRs with slow radial propagations would probably belong to principal bands. However, these quasi-stationary rainbands are quite few (only ~14%), and most of the outer TCRs analyzed in this study are characterized by radially propagating nature. In this revision, this part of the text has been reworded to make this point clear. What’s radial distances for these slow-propagating rainbands?

Line 355: Add citation to support use of $n=2$ for tangential wave number.

Reply: Supportive references have been cited in this revision.

Line 393: Suggesting changing “the squall-line characteristics” to “the above outlined squall-line characteristics.”

Reply: Reworded as suggested.

Line 395: Please define “low-level,” “low-level band-normal vertical shear” and “band-relative band-normal wind component.” What levels and distances are used? How are the quantities calculated? Information can also be added to the Figure 7 caption.

Reply: Height information for the calculation of low-level vertical shear and band-

normal wind component has been added in the caption of Fig. 7.

Line 420-421: This is where I really started to get wary of your use of ERA-5 data for these mesoscale circulations associated with cold pool dynamics.

Reply: Please refer to our responses to Major comment#1.

Line 428-429: This is a good way to account for some of the uncertainty. Unfortunately, due to data quality issues, I think the uncertainty is much larger.

Reply: Please refer to our responses to Major comment#1.

Line 434-444: “that there is a much higher percentage of the squall line TCR group compared to that of the non-squall-line TCR group.” I don't follow, do you mean a higher percentage of overlap? or do you mean the density of the lines in the plot? like you've squashed the data more in Figure 7c so that more cases fit into the plot?

Reply: Yes, we mean a much higher percentage of agreement between the observed propagation velocities and the theoretically estimated cold pool velocities for the squall-line TCR group compared to that of the non-squall-line TCR group. This part of the text has been reworded for clarity.

Line 448: characters • characteristics

Reply: Revised as suggested.

Line 448: “fast movement” For clarity, are you referring to faster motion (blue dots higher magnitude) in Figure 7c?

Reply: The “fast movement” described here is referring to squall-line convective systems that typically have faster propagations. Relevant articles are also cited, along with this statement.

Line 448-450: “One might see a more rapidly propagating nature for convective systems if they evolve with the operation of squall-line dynamics.” More rapid compared with what?

Reply: This sentence has been reworded for clarity.

Line 490: When you refer to the rainbands being inside versus outside the “storm size,” I don’t think “storm size” is the word you want since you can’t really be inside the size. Rather, the rainbands are in/outside the storm radius, storm perimeter, or potentially the storm area.

Reply: Thank the Reviewer for raising the valid point. In this revision, we explicitly define R34 as “storm radius”, instead of simply using the term “TC size”, to avoid any possible ambiguity. For clarity, this part of the text has also been reworded in this revision.

Line 512: “must have existed” • must also exist

Reply: Reworded as suggested.

Line 540-546: Please be sure that you are providing information about all the datasets you used in this study, including the JTWC best track data.

Reply: Yes, the website information for the JTWC best track data has been added in the Data availability section.

Figure 1: I am wondering if there is a better way than arrows to demarcate the rainband. Could you reduce the transparency of the rest of the radar reflectivity regions? so that the rainband stands out more.

Reply: Following the reviewer’s suggestion, Figure 2 has been improved to highlight the rainband by reducing the transparency of the rest of the radar reflectivities. In addition, after we remove the arrows, the rainband position is not very easy to identify visually, especially during the preformation stage (i.e., Fig. 2a). Thus, we also choose to include the arrows in this revision for clarity.

Figure 2: Could you add percentages or fraction on the right axis to assist with interpretation? This would be especially helpful with comparing fraction of inside vs. outside TC rainbands cited in the text.

Reply: After adding percentages on the right axis, we found that it makes the figure illustration very busy and that the digits of percentage are difficult to read especially for the area closer to the inner core of TCs, where many rainbands form. So, we

choose not to add the percentages on the figure. However, alternative information can be provided by Fig. 4a, which shows the number of TCR cases formed at different intervals of radial distances.

Figure 5a,b,d: For interpretation, I manually added vertical lines along $x=0$ so that I could distinguish positive and negative values more easily. I think it would be great to add these to the plots.

Reply: Vertical lines at $x=0$ have been added in these plots, as the Reviewer suggested.

REVIEWER COMMENTS

Reviewer #1 (Remarks to the Author):

Summary:

The authors revisions have substantially improved the manuscript, which presents a very nice radar climatology of outer-core tropical cyclone rainbands from tropical cyclones near Taiwan from 2002-2019. In particular, the addition of Figure 8 clearly highlights the importance of pre-existing precipitation in the vicinity of the region where the outer TC rainband later develops. This preexisting precipitation is a likely source of cold pools which may help to initiate the outer TC rainband. I also commend the authors for the addition of the schematic in Figure 9 which nicely summarizes both squall-line and non-squall-line outer TC rainband development.

Minor comments:

L45-47: The addition of this sentence I believe strengthens the abstract greatly and makes the understanding of the outer TC rainband origin much clearer.

L60: "the prevention and prediction" should be changed to "disaster mitigation and the prediction". It is not possible to prevent the natural hazard (the TC) but it may be possible to mitigate some of the impacts.

L69: I recommend the authors remove the comparison with inner TCR impacts here. For example, remove "In contrast to inner TCRs" and instead say "Outer TCRs tend to have strong and broad impacts...".

L97-103: I think it is important within this section to clearly state that outer TCRs are a broader class of rainbands than just a principal rainband that extends into this region.

L139: "hypothetical" should be changed to "uncertain".

L186: "calculated to obtain" can be rewritten as "used to calculate".

L327-334: I appreciate the authors verifying that there is no diurnal variation in outer TCR formation. This is a noteworthy and useful result.

L332-333: I think "although the peak and lowest..." portion of this sentence can be removed because there is no clear diurnal signal anyway.

Figure 8: I suggest adding grid lines to this figure to make the distance relative to the origin location easier to see.

L604-605: “would be probably” should be changed to “could be”.

L641-644: This sentence is very clear and gets at the importance of cold pools in initiating many of the outer TC rainbands.

Reviewer #2 (Remarks to the Author):

Review of “Origin of outer tropical cyclone rainbands”

Synopsis:

The authors have addressed most of my concerns from the first review. However, I have a few remaining concerns that need to be addressed before I can recommend this study for publication.

Major comments:

1. Regarding the use of ERA5 data, thank you for taking the time to verify that the winds are similar to observations. In this latest revision, though, I am left with some confusion about the wind data that was used throughout the study. What vertical levels are chosen? and for which parts of the analysis? On lines 430-454, the authors discuss the verification that they completed to ensure that the ERA5 winds are consistent with observations. However, they do not specify the vertical level that they use in this comparison. Is it the 1.5 km MSL that you use in the filamentation time calculation? Please specify the vertical levels that are used in the following places:

- a. What vertical level(s) are used in the verification on lines 430-454
- b. What vertical level(s) are used in the filamentation time calculation (here, I think that the wind information is specified more clearly in the text, but I include it to emphasize that the authors are using ERA5 wind data in multiple places)
- c. ERA5 wind data are also used in the VRW calculations, with the mean tangential wind being averaged “within the layer of precipitation associated with outer TCRs (0.1-12 km MSL). How do you justify using winds within the boundary layer in this calculation?
- d. What vertical level(s) are used in the squall line dynamic section, i.e., low-level vertical shear and band-normal wind calculations. In the Figure 7a caption, the authors describe the low-level vertical wind shear as being “calculated and averaged below 1.5-2 MSL.” Does this include all ERA5 vertical levels below 1.5-2 km above MSL? If so, then the authors are also including wind data within the boundary layer, which might be problematic. This might be specified on line 524, but it’s not clearly stated in my opinion.
- e. Are there other places that I have missed?

I am asking that the authors provide a summary of the ERA5 data used in the study, and some justification for its use throughout. If boundary layer winds are being used, is this justified and how?

2. Thank you for providing additional details about the subjective screening of radar data to locate outer TC rainbands. This is helpful. The additional details about the spatial measurements and identification of orographically generated rainbands, etc. are also welcome and provide important details about your methods.

a. I think that it would be possible to remove principal rainbands from this study, since these rainbands are large, prominent, and persistent, and located downshear/downshear left. I believe that the origin mechanisms for outer TCRs and PBs are likely to be different and the inclusion of PBs in this study is likely “contaminating” the results. Please provide some additional discussion of this fact. I think this should be considered a limitation in the conclusions section.

Minor comments:

Line 185: remove the word “just”

Line 186: Rather than, “Reflectivity data ... are then calculated”, I think you mean “Reflectivity data ... are used”

Lines 187-207: Why are 20 dBZ and 30 dBZ threshold selected in this section. Please elaborate and/or provide justification.

NCOMMS-23-16082A
Responses to Reviewer#1

Reviewer #1 (Remarks to the Author):

Summary:

The authors revisions have substantially improved the manuscript, which presents a very nice radar climatology of outer-core tropical cyclone rainbands from tropical cyclones near Taiwan from 2002-2019. In particular, the addition of Figure 8 clearly highlights the importance of pre-existing precipitation in the vicinity of the region where the outer TC rainband later develops. This preexisting precipitation is a likely source of cold pools which may help to initiate the outer TC rainband. I also commend the authors for the addition of the schematic in Figure 9 which nicely summarizes both squall-line and non-squall-line outer TC rainband development.

Reply: We sincerely appreciate Reviewer#1's positive and encouraging comments on our revision. Thank you so much. A set of responses to your further comments is provided below.

Minor comments:

L45-47: The addition of this sentence I believe strengthens the abstract greatly and makes the understanding of the outer TC rainband origin much clearer.

Reply: Thank you very much.

L60: “the prevention and prediction” should be changed to “disaster mitigation and the prediction” . It is not possible to prevent the natural hazard (the TC) but it may be possible to mitigate some of the impacts.

Reply: Thank Reviewer#1 for the good point. This sentence has been revised as suggested.

L69: I recommend the authors remove the comparison with inner TCR impacts here. For example, remove “In contrast to inner TCRs” and instead say “Outer TCRs tend to have strong and broad impacts…” .

Reply: This is a valid point. This sentence has been revised as suggested.

L97-103: I think it is important within this section to clearly state that outer TCRs are a broader class of rainbands than just a principal rainband that extends into this region.

Reply: Thank Reviewer#1 for the good point. We have clearly stated this point in this paragraph for clarity.

L139: “hypothetical” should be changed to “uncertain” .

Reply: Revised as suggested.

L186: “calculated to obtain” can be rewritten as “used to calculate” .

Reply: Revised as suggested.

L327-334: I appreciate the authors verifying that there is no diurnal variation in outer TCR formation. This is a noteworthy and useful result.

Reply: Thank you.

L332-333: I think “although the peak and lowest...” portion of this sentence can be removed because there is no clear diurnal signal anyway.

Reply: Removed as suggested.

Figure 8: I suggest adding grid lines to this figure to make the distance relative to the origin location easier to see.

Reply: In this revision, grid lines have been added in Fig. 8 as suggested.

L604-605: “would be probably” should be changed to “could be” .

Reply: Revised as suggested.

L641-644: This sentence is very clear and gets at the importance of cold pools in

initiating many of the outer TC rainbands.

Reply: Thank you very much.

NCOMMS-23-16082A
Responses to Reviewer#2

Reviewer #2 (Remarks to the Author):

Review of “Origin of outer tropical cyclone rainbands”

Synopsis:

The authors have addressed most of my concerns from the first review. However, I have a few remaining concerns that need to be addressed before I can recommend this study for publication.

Reply: We appreciate Reviewer#2’s positive and further comments, which help us improve the manuscript. Thank you very much. A set of responses to your detailed comments is provided below.

Major comments:

1. Regarding the use of ERA5 data, thank you for taking the time to verify that the winds are similar to observations. In this latest revision, though, I am left with some confusion about the wind data that was used throughout the study. What vertical levels are chosen? and for which parts of the analysis? On lines 430-454, the authors discuss the verification that they completed to ensure that the ERA5 winds are consistent with observations. However, they do not specify the vertical level that they use in this comparison. Is it the 1.5 km MSL that you use in the filamentation time calculation? Please specify the vertical levels that are used in the following places:
 - a. What vertical level(s) are used in the verification on lines 430-454
 - b. What vertical level(s) are used in the filamentation time calculation (here, I think that the wind information is specified more clearly in the text, but I include it to emphasize that the authors are using ERA5 wind data in multiple places)
 - c. ERA5 wind data are also used in the VRW calculations, with the mean tangential wind being averaged “within the layer of precipitation associated with outer TCRs (0.1-12 km MSL). How do you justify using winds within the boundary layer in this calculation?
 - d. What vertical level(s) are used in the squall line dynamic section, i.e., low-level vertical shear and band-normal wind calculations. In the Figure 7a caption, the authors describe the low-level vertical wind shear as being “calculated and averaged below 1.5-2 MSL.” Does this include all ERA5 vertical levels below 1.5 -2

km above MSL? If so, then the authors are also including wind data within the boundary layer, which might be problematic. This might be specified on line 524, but it's not clearly stated in my opinion.

e. Are there other places that I have missed?

I am asking that the authors provide a summary of the ERA5 data used in the study, and some justification for its use throughout. If boundary layer winds are being used, is this justified and how?

Reply: We thank you for raising the issue of what vertical levels of ERA5 data are used in this study. Below is a summary of vertical levels used for different parts of our analysis and their justification. Additional descriptions/explanations have been given in appropriate places of this revised manuscript for clarity.

- a. Verification of ERA5 wind data: Surface observations from eight offshore island stations represent “ground truth” measurements and are used to evaluate the reliability of ERA5 wind data. For a meaningful comparison with surface observations, the vertical level of the ERA5 wind data at 10-m (MSL) (i.e., near-surface winds) is chosen for this verification. This information has been stated clearly in the Propagation and geometry characteristics section. As elaborated in the text of the revised manuscript and previous responses, these comparisons show general consistency between island surface observations and ERA5 data, which in turn suggests that ERA5 can provide generally reliable boundary-layer wind information in the environment of outer TCRs.
- b. Filamentation time calculation: In this study, the vertical level of the ERA5 wind data at 1.5 km (MSL) is chosen for the filamentation time calculation. It has been recognized that most intense winds associated with TCs usually occur immediately above the top of the boundary layer. Given a typical height of TC boundary layer at around 0.5-1 km MSL (e.g., Zhang et al. 2011, <https://doi.org/10.1175/MWR-D-10-05017.1>), the wind data from the vertical level used here are expected to better reflect the TC intensity and circulation so their calculated results for the filamentation time could be more representative. These explanations have been added in the Formative location and dynamic regime section of this revision.
- c. VRW calculations: The vertical levels of the ERA5 wind data from 0.1 to 12 km MSL are used to calculate the mean tangential wind expressed in Eqs. (5) and (6) for the VRW calculations. The tangential vertical shear in Eq. (7) is approximated by the difference in the tangential wind between 0.1 and 12 km MSL. As described in the Propagation and geometry characteristics section of the original

and revised manuscript, the vertical wavelength of both GWs and VRWs in this study is assumed to be the typical depth of convective motions associated with outer TCRs extending from the oceanic surface to ~12 km MSL (Yu et al. 2018²⁹). The deep layer chosen for the VRW calculations is corresponding to the characteristic scale of wave motions. In addition, the reason why we choose the 0.1 km MSL as the lowest level in these calculations, instead of simply using the near-surface level, is that this level is well above the top of surface layer so the uncertainty on winds due to strong and direct impact of surface friction can be appropriately eliminated. The explanations above have been added in this revision. It should be noted that the theoretically calculated VRW phase velocities are not sensitive to the different choices of the lowest level. Specifically, the changes in the calculated radial and tangential phase velocities of VRWs for all studied outer TCR cases are found to be within 0.1 and 1.0 m s⁻¹, respectively, when using different heights of the lowest level from the 0.1 km MSL to a height above the top of the boundary layer (1.5 km MSL).

- d. Evaluation of squall-line dynamics: In this study, the low-level mean band-normal vertical shear is approximated by the difference in the band-normal wind between 0.1 and 2 km MSL. This layer is corresponding to the mean depth of cold pools associated with outer TCRs, as revealed by previous studies of TCs²⁸⁻³⁰. Similar to the consideration for the VRW calculations as described above, choosing the lowest level at 0.1 km is to preclude the direct influences of surface friction. In addition, the reason why the mean band-normal wind is calculated below 1.5 km MSL is because this layer is corresponding to the mean depth of low-level inflow (i.e., front-to-rear flow) feeding the leading edge of the outer TCRs documented in this study. These explanations have been added in the Squall-line dynamics section of this revision.
- e. Please see our responses to Major comment#1c above for details.

2. Thank you for providing additional details about the subjective screening of radar data to locate outer TC rainbands. This is helpful. The additional details about the spatial measurements and identification of orographically generated rainbands, etc. are also welcome and provide important details about your methods.

a. I think that it would be possible to remove principal rainbands from this study, since these rainbands are large, prominent, and persistent, and located downshear/downshear left. I believe that the origin mechanisms for outer TCRs and PBs are likely to be different and the inclusion of PBs in this study is likely “contaminating” the results. Please provide some additional discussion of this fact. I think this should be considered a limitation in the conclusions section.

Reply: We thank you for pointing out the possible inclusion of PBs. In this study, a precipitation feature is identified as outer TCRs as long as it satisfies several objective, quantitative criteria, as described in the Data and rainband identification section of the manuscript. The propagation characteristics and formative locations of these identified rainbands may provide some helpful hints on the degree to which the outer TCR dataset analyzed in this study contain PBs. One of the most important characteristics for PBs are their quasi-stationary nature relative to the TC center. PBs also tend to be located downshear (as Reviewer kindly mentioned) and near the outer boundary of the inner-core vortex of TCs. It is found that there are small proportions of the identified outer TCRs (144 TCR cases, ~14%) that exhibit slow radial propagations within $\pm 1 \text{ m s}^{-1}$ (Fig. 5a) and are consistent with quasi-stationary nature. Statistical analyses for the locations of these quasi-stationary cases (i.e., the PB candidates) further indicate that they are distributed over different quadrants with respect to the large-scale shear vector (Fig. R1) and over a broad range of radial distance (Fig. R2), with a mean of 331 km and 10.5 times RMW. Assuming the 2-5 times RMW as an approximate area defining the outer boundary of the inner-core vortex of TCs, only 13 out of 144 outer TCR cases are actually located in the large-scale downshear quadrants within this area. Such a small number of rainbands with PB characteristics (~1% of all outer TCRs identified in this study) suggests that the possible impact of PBs on the outer TCR dataset and on the statistical results presented in this article may be considered negligible.

One of the primary factors contributing to the potential preclusion from the class of outer TCRs is related to the fact that PBs tend to occur preferentially inside the TC vortex core or near its outer boundary (~3 times RMW), as learned from the detailed examination of radar-observed precipitation field from 95 TCs analyzed in this study. An example of TCRs observed from Typhoon Linfa (2015) is chosen to illustrate this common situation, as shown in Fig. R3. In this case, one PB-like rainband and several outer TCRs, located in the northern and eastern portion of the typhoon, respectively, are co-existing but they are well separate from each other. The rainband segment for the PB-like rainband is mostly located well within and close to the ring of 3 times RMW, although its upwind segment is located slightly beyond 3 times RMW. In contrast, the observed outer TCRs are primarily located in outer regions of the typhoon. The inner PB-like rainband shown in Fig. R3 is not included in our outer TCR dataset not only because its initiation scenario is not well captured by radar observations but also because it does not satisfy the criterion of radial distance (i.e., > 3 times RMW) used to identify outer TCRs.

It is noteworthy that whether PBs belong to inner rainbands, outer rainbands or some sort of their mixed behavior still remains ambiguous in the literature due to the lack of large collection of both inner/outer TCRs and PB cases and their relevant statistical analyses and comparisons. To clarify this scientific issue deserves future exploration but will be very challenging because it is usually difficult for atmospheric observational platforms to provide complete and continuous pictures of TC precipitation over the open ocean. Complicated organization and rapidly evolving nature of TC precipitation also represents an additional challenge for the investigation of this topic.

We thank you very much again for bringing the PB issue to our attention. In response to the Reviewer's comment, discussions on the possible inclusion of PBs as described above have been added in the Propagation and geometry characteristics section of this revised manuscript for clarity.

Fig. R1. (a) Plan view of formative locations for 144 quasi-stationary outer TCR cases relative to the TC center and large-scale deep-layer vertical shear. The large-scale shear is calculated within the deep layer between 700 and 200 hPa over a radial distance of 150-500 km from the ERA5 reanalysis data. (b) Same as (a), but showing the formative locations as a function of normalized radial distance by RMW.

Fig. R2. Formative number of 144 quasi-stationary outer TCR cases at different intervals of (a) radial distance (km) and (b) normalized radial distance by RMW. In (b), the number of outer TCR cases located beyond 15 times RMW (20 cases) are not shown.

Fig. R3. Low-level composite radar reflectivity (dBZ) from Typhoon Linfa valid at 0030 UTC on 8 July 2015. The dashed circle denotes the inner-core boundary (3 times RMW) of the typhoon. The principal band discussed in the text of our responses is denoted by PB. The solid ellipse encompasses the primary region with outer TCR activities.

Minor comments:

Line 185: remove the word “just”

Reply: Removed as suggested.

Line 186: Rather than, “Reflectivity data ... are then calculated” , I think you mean “Reflectivity data ... are used”

Reply: This sentence has been revised as suggested for clarity.

Lines 187-207: Why are 20 dBZ and 30 dBZ threshold selected in this section. Please elaborate and/or provide justification.

Reply: We have provided justification within this paragraph for the 20 and 30 dBZ threshold used in this study.

REVIEWERS' COMMENTS

Reviewer #2 (Remarks to the Author):

Review of "Origin of outer tropical cyclone rainbands"

Synopsis:

The authors have addressed my concerns, and I am happy to recommend this study for publication.

Minor comments:

Line 194: Suggest further justification for using 20 and 30 dBZ. In particular, the justification for 30 dBZ is flimsy and seems to be somewhat arbitrary.

Line 197: missing the word "is": "The reason why a lower reflectivity threshold of 20 dBZ IS used here is because..."

NCOMMS-23-16082B
Responses to Reviewer#2

Reviewer #2 (Remarks to the Author):

Review of “Origin of outer tropical cyclone rainbands”

Synopsis:

The authors have addressed my concerns, and I am happy to recommend this study for publication.

Reply: The authors appreciate Reviewer#2's positive and encouraging comments on our revised manuscript. Thank you so much.

Minor comments:

Line 194: Suggest further justification for using 20 and 30 dBZ. In particular, the justification for 30 dBZ is flimsy and seems to be somewhat arbitrary.

Reply: Following Reviewer's suggestion, further justification for using 20 and 30 dBZ has been provided in the Methods section of the revised manuscript.

Line 197: missing the word “is” : “The reason why a lower reflectivity threshold of 20 dBZ IS used here is because...”

Reply: Corrected as suggested.